# Dual Risk Minimization: Towards Next-Level Robustness in Fine-tuning Zero-Shot Models

**Kaican Li**[1*]  **Weiyan Xie**[1*]  **Yongxiang Huang**[2]  **Didan Deng**[2]  **Lanqing Hong**[2]
**Zhenguo Li**[1,2]  **Ricardo Silva**[3]  **Nevin L. Zhang**[1†]

[1]The Hong Kong University of Science and Technology
[2]Huawei    [3]University College London

## Abstract

Fine-tuning foundation models often compromises their robustness to distribution shifts. To remedy this, most robust fine-tuning methods aim to preserve the pre-trained features. However, not all pre-trained features are robust and those methods are largely indifferent to which ones to preserve. We propose dual risk minimization (DRM), which combines empirical risk minimization with worst-case risk minimization, to better preserve the core features of downstream tasks. In particular, we utilize core-feature descriptions generated by LLMs to induce core-based zero-shot predictions which then serve as proxies to estimate the worst-case risk. DRM balances two crucial aspects of model robustness: expected performance and worst-case performance, establishing a new state of the art on various real-world benchmarks. DRM significantly improves the out-of-distribution performance of CLIP ViT-L/14@336 on ImageNet ($75.9 \rightarrow 77.1$), WILDS-iWildCam ($47.1 \rightarrow 51.8$), and WILDS-FMoW ($50.7 \rightarrow 53.1$); opening up new avenues for robust fine-tuning. Our code is available at `https://github.com/vaynexie/DRM`.

## 1 Introduction

Foundation models such as CLIP (Radford et al., 2021) and ALIGN (Jia et al., 2021) have revolutionized machine learning with their remarkable zero-shot and adaptive capabilities. Research has shown that such capabilities are mainly due to robust feature representations gained from large-scale training data (Fang et al., 2022; Xu et al., 2024). The models have been proven useful in various downstream tasks (Shen et al., 2022; Zhang et al., 2022; Betker et al., 2023; Pi et al., 2024) and are the cornerstones of large multimodal models (Alayrac et al., 2022; Liu et al., 2023; Zhu et al., 2024).

*Fine-tuning* is one of the most common approaches to the downstream adaptation of foundation models (Bommasani et al., 2021; Shen et al., 2022). However, such adaptation often comes at the cost of robustness (Radford et al., 2021; Pham et al., 2023), resulting in larger gaps between downstream in-distribution (ID) and out-of-distribution (OOD) performance (Wortsman et al., 2022).

Kumar et al. (2022) showed that fine-tuning tends to distort pre-trained features, and the distortion is exacerbated by randomly initialized heads which would significantly alter the pre-trained features to fit ID examples. The proposed remedy, LP-FT, first learns a linear probe (LP) on frozen features, and then followed by regular fine-tuning (FT). Goyal et al. (2023) took this idea further by reusing the pre-trained text encoder of CLIP as the classification head for fine-tuning. This method improves LP-FT and is colloquially known as "fine-tune like you pre-train" (FLYP). WiSE-FT (Wortsman et al., 2022) investigated combining pre-trained models with their fine-tuned versions by weight averaging, which can be seen as yet another approach to recover robust features lost during fine-tuning.

---

[*] Equal contribution, listed in alphabetical order.
[†] Correspondence to lzhang@cse.ust.hk.

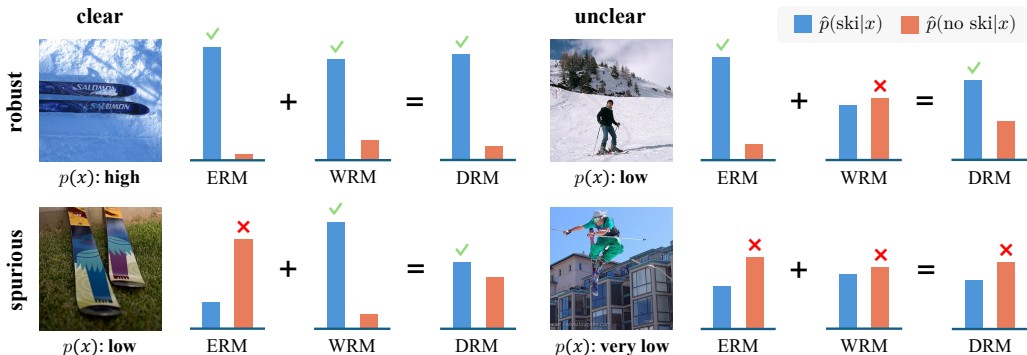

Figure 1: **Dual risk minimization (DRM) combines empirical risk minimization (ERM) and worst-case risk minimization (WRM) to complement their weaknesses.** In this simple binary classification task predicting if there are skis in a given image, (i) ERM underperforms when the core features (the appearance of ski) are *clear* but the non-core features such as background/context are *spurious* (i.e. negatively correlated with ski), and (ii) WRM underperforms when the core features are *unclear* but the non-core features are *robust* (i.e. positively correlated with ski). DRM outperforms ERM and WRM under mild conditions such that the core features are not always clear and the non-core features are more often robust than not.

While the existing approaches aim to preserve pre-trained features, the fine-tuning process is still guided by empirical risk minimization (ERM; Vapnik, 1998), which favors the most predictive but not necessarily the most robust features. In general, there are two kinds of robust features: *core features* which essentially define the target classes, and *non-core features* that may aid prediction when the core features are not *clear* (Gao et al., 2023). ERM models tend to exploit the non-core features even when the core features are clear (Geirhos et al., 2020; Shah et al., 2020). This often harms OOD performance as non-core features are generally less reliable out-of-distribution.

To better preserve the core features, we propose a new principle called *dual risk minimization* (DRM) which combines ERM with worst-case risk minimization (WRM; Wald, 1945), a common principle for domain generalization (Arjovsky et al., 2019; Sagawa et al., 2020; Cha et al., 2021; Kirichenko et al., 2023). This combination rests on our view that robustness involves *two* main aspects: the *expected* (or average) performance and the *worst-case* performance over all domains. While there is often a trade-off between these two aspects (Tsipras et al., 2019; Teney et al., 2023), Figure 1 illustrates how DRM balances the trade-off to improve overall robustness.

The main challenge of applying DRM to real-world tasks is to assess of worst-case risk. To this end, we use *concept descriptions* (Pratt et al., 2023)—short texts that describe the core features of each class—obtained with GPT-4 (Achiam et al., 2023). The description for *cougar*, for instance, is "*a large, tawny cat with a muscular build and a small head.*" We feed these descriptions to a pre-trained CLIP text encoder (Radford et al., 2021) for the text embeddings, which are then used to construct soft class labels for each training image according to the similarity scores between the image and text embeddings. The risk w.r.t. the soft labels can be seen as a proxy of the worst-case risk and is thus minimized instead. Empirically, DRM significantly outperforms the state of the art on challenging benchmarks such as ImageNet (Deng et al., 2009) and WILDS (Koh et al., 2021).

In summary, we make the following key contributions in this paper:

- We propose *dual risk minimization* (DRM), a novel approach that combines ERM and WRM to improve downstream robustness of zero-shot foundation models while addressing the intractability of WRM through innovative use of concept descriptions.

- We highlight that robustness for many real-world problems concerns both *expected* and *worst-case* performance while most previous works focus on only one. We then show that DRM offers a simple and effective way to balance these two important aspects of robustness.

- We establish a strong new state of the art on multiple real-world benchmarks, promising next-level robustness in fine-tuning zero-shot models. On CLIP ViT-L/14@336, DRM achieves a significant, over 5% relative improvement in OOD performance over the best baseline method.

## 2 Related work

**Robust fine-tuning of pre-trained models.** Prior to the work of Kumar et al. (2022); Wortsman et al. (2022); Goyal et al. (2023) which we have introduced, Li et al. (2018) proposed to restrict the $L^2$ distance between the parameters of pre-trained and fine-tuned models via regularization. Some other work explored updating only a small number of (pre-trained/add-on) parameters (Guo et al., 2019; Zhang et al., 2020; Gao et al., 2024b). Similar ideas (Kirkpatrick et al., 2017; Zenke et al., 2017) were also discussed in continual learning to mitigate catastrophic forgetting (McCloskey and Cohen, 1989). Apart from explicit constraints on model parameters, Ge and Yu (2017) turned to the source of robust features and proposed to incorporate a subset of pre-trained data for fine-tuning, while Cha et al. (2022) aimed to enhance the mutual information between pre-trained and fine-tuned features. Jiang et al. (2019); Zhu et al. (2020) added smoothness constraints on model predictions for adversarial examples (Szegedy et al., 2013) to help retain robust features. Andreassen et al. (2021) showed that OOD accuracy tends to improve initially but then plateaus as the fine-tuning proceeds. For more discussion on related work including concurrent ones, see Appendix B.

**Worst-case risk minimization.** The study of worst-case risk minimization (WRM) dates back to the work of Wald (1945), which has gradually evolved into what we know as robust optimization today (Ben-Tal et al., 2009). More recently, WRM has been considered (by many) a basic principle for domain generalization (DG; Blanchard et al., 2011; Muandet et al., 2013). A notable example is invariant risk minimization (IRM; Arjovsky et al., 2019), which aims to learn core-feature representations from multi-domain data. Such representations, under mild causal assumptions, give rise to classifiers that minimize the worst risk (Peters et al., 2016). Another key method, GroupDRO (Sagawa et al., 2020), imposes higher penalties on domains with higher empirical risks. Unlike DRM, neither IRM nor GroupDRO formulates WRM as an explicit optimization constraint for ERM. Eastwood et al. (2022) pointed out that sacrificing too much average performance for worst-case performance is not ideal for DG. Hence, they proposed to minimize the risk among the most likely domains. Lastly, our setup is partly similar to Alabdulmohsin et al. (2023) which also relies on external information.

**Prompt design for zero-shot classification.** To better leverage the capability of zero-shot models, various prompt designs have been proposed. Menon and Vondrick (2022); Pratt et al. (2023); Maniparambil et al. (2023) mainly explored prompts for zero-shot classification. Their prompts were generated by LLMs (Radford et al., 2019) with slightly different instructions than ours, not explicitly focusing on core features. For example, Pratt et al. (2023) used "*Describe an image from the internet of a(n) ...*", which may inadvertently introduce descriptions of non-core features in the resulting prompts. The prompts considered by Yang et al. (2023); Yan et al. (2023) are closer to ours in this respect, where they used LLM-generated concept descriptions to build concept bottleneck models for interpretable image classification. More recently, Mao et al. (2024) proposed to use context-aware prompts such as "*a [context] of [class name]*," while Cheng et al. (2024) used both domain-invariant and domain-specific prompts generated by LLMs. However, both methods require either image context or specific domain information to generate the prompts.

## 3 Dual risk minimization

**Data model.** Let $X$ and $Y$ be the input and *ground-truth* target variables for which we adopt the following data generation model:

$$
\begin{aligned}
X &\leftarrow h_X(X_\text{c}, X_\text{n}, \varepsilon), \\
Y &\leftarrow h_Y(X_\text{c});
\end{aligned}
\tag{1}
$$

where $(X_\text{c}, X_\text{n})$ are latent variables and $\varepsilon$ is exogenous noise. We call $X_\text{c}$ *core features* and $X_\text{n}$ *non-core features* of $(X, Y)$. $X_\text{n}$ and $Y$ may be correlated due to hidden confounders of $(X_\text{c}, X_\text{n})$ and direct causal mechanisms between $(X_\text{c}, X_\text{n})$. Following Peters et al. (2016), we assume the causal mechanisms and the distribution of $\varepsilon$ are invariant across domains. There are no other hidden variables or mechanisms. Similar models were widely adopted in the literature (Tenenbaum and Freeman, 1996; Mahajan et al., 2021; Mitrovic et al., 2021; Ahuja et al., 2021; Liu et al., 2021; Lv et al., 2022; Ye et al., 2022; Zhang et al., 2023a; Gao et al., 2024a) where $X_\text{c}$ and $X_\text{n}$ are sometimes referred to as 'content' and 'style'. We use calligraphic letters such as $\mathcal{X}$ and $\mathcal{Y}$ to denote the set of possible outcomes of the random variables.

**Ideal objective for robustness.** Let $\mathcal{D}$ be all possible domains of a task, and $\mathscr{P}$ be some natural distribution over $\mathcal{D}$. By definition, $\mathscr{P}(d) > 0$ for all $d \in \mathcal{D}$. Every domain $d$ is associated with a data distribution $p_d(x, y, x_c, x_n)$ consistent with (1). Let $\hat{p}_\theta(y|x)$ be a prediction model parameterized by $\theta \in \Theta$. Its risk in terms of negative log-likelihood,

$$R_d(\theta) = \mathbb{E}_{(x,y) \sim p_d}[-\log \hat{p}_\theta(y|x)], \tag{2}$$

can be seen as a measure of its performance in domain $d$. Let $d_s \in \mathcal{D}$ be the training domain. For simplicity, we will omit $d$ when it is clear from the context, e.g., $R_{d_s}(\theta)$ will be written as $R_s(\theta)$.

For real-world applications, we argue that a *robust* model should optimize its *expected* performance over $\mathscr{P}$ while maintaining acceptable *worst-case* performance across $\mathcal{D}$. The expected performance implies how well the model would perform at the most general population level, while the worst-case performance tells us the model's performance in the worst scenario one may encounter. We note that Eastwood et al. (2022) and Zhang et al. (2023b) share a similar view with us on robustness.

We formalize the above intuition as the following constrained optimization problem, namely *idealized dual risk minimization* (IDRM), which aims to minimize the empirical risk of $\hat{p}_\theta(y|x)$ while ensuring its worst-case risk is below some threshold value $\alpha$:

$$\min_{\theta \in \Theta} R_s(\theta) \quad \text{subject to} \quad \max_{d \in \mathcal{D}} R_d(\theta) \le \alpha. \tag{IDRM}$$

IDRM generalizes ERM (Vapnik, 1998) and WRM (Wald, 1945) as it reduces to ERM when $\alpha$ is large and to WRM when $\alpha$ is small. IDRM also bears some resemblance to IRM (Arjovsky et al., 2019), which involves an implicit WRM constraint. The constraint, however, requires the classification head to be *optimal* in all training domains and thus may be too demanding in practice. Another closely related work, GroupDRO (Sagawa et al., 2020), proposes to minimize the worst training-domain risk—a more empirical flavor of WRM. Both IRM and GroupDRO rely on ideally grouped training data to capture invariance across domains. In Section 4, we will show that this is largely unnecessary for zero-shot models and provide a practical solution for IDRM with just *single-domain* data.

**From IDRM to DRM.** IDRM can be solved as the following unconstrained optimization problem due to strong duality (proof in Appendix A).

**Theorem 1.** *Strong duality holds between IDRM and the following dual problem:*

$$\max_{\lambda' \ge 0} \min_{\theta \in \Theta} \left[ R_s(\theta) + \lambda' \max_{d \in \mathcal{D}} R_d(\theta) \right] - \lambda'\alpha. \tag{3}$$

Let $\lambda^\star$ be any solution of $\lambda'$ to (3). By the strong duality, IDRM then reduces to $\min_{\theta \in \Theta}[R_s(\theta) + \lambda^\star \max_{d \in \mathcal{D}} R_d(\theta)]$. The worst-case risk, i.e., $\max_{d \in \mathcal{D}} R_d(\theta)$, is still intractable in itself, but it is closely related to the degree to which the model $\hat{p}_\theta(y|x)$ relies on core features to predict $y$. This is because for a diverse set of domains $\mathcal{D}$, leveraging non-core features would always lead to worse performance in certain domains (Arjovsky et al., 2019; Geirhos et al., 2020). To minimize the worst-case risk, therefore, the model must only utilize the core features to make prediction.

Suppose there is an oracle feature extractor $f_c$ that returns a faithful representation of the core features of any input $x$. As the core features may not always be clear, $f_c(x)$ can be viewed as some distribution over the core features for each $x$. Let $p_c(y|x)$ be the optimal model that can be built upon $f_c(x)$. The risk of $\hat{p}_\theta(y|x)$ w.r.t. $p_c(y|x)$ on the training domain $d_s$ is given by

$$R_s^c(\theta) = \mathbb{E}_{x \sim p_s} \mathbb{E}_{y \sim p_c(y|x)}[-\log \hat{p}_\theta(y|x)]. \tag{4}$$

Assuming $p_s(x)$ is fairly diverse, the risk $R_s^c(\theta)$ measures the degree to which the model's prediction is based on the core features and thus can be viewed as a proxy for the worst-case risk. Hence, we can replace $\max_{d \in \mathcal{D}} R_d(\theta)$ with $R_s^c(\theta)$ while still achieving a similar optimization effect.

In summary, we relax IDRM to the following DRM formulation:

$$\min_{\theta \in \Theta} R_s(\theta) + \lambda R_s^c(\theta) \tag{DRM}$$

with some properly chosen $\lambda \ge 0$. Now the risk $R_s^c(\theta)$ can also be interpreted as a regularization term for ERM to help preserve the core features. In the following section, we demonstrate how to utilize zero-shot models like CLIP models (Radford et al., 2021) to estimate $p_c(y|x)$, and subsequently how to apply DRM to robustly fine-tune the same CLIP models.

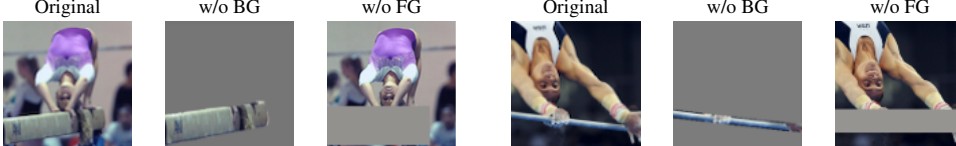

| | Original | w/o BG | w/o FG | Original | w/o BG | w/o FG |
|---|---|---|---|---|---|---|
| df | An image of balance beam. | | | An image of gymnastic horizontal bar. | | |
| cd | A long, thin piece of wood or metal elevated off the ground. | | | Long metal or wood bar held up by upright supports. | | |
| df | 0.367 | 0.262 (−28.6%) | 0.395 ( +7.6%) | 0.333 | 0.267 (−19.8%) | 0.367 (+10.2%) |
| cd | 0.286 | 0.281 ( −1.7%) | 0.123 (−57.0%) | 0.258 | 0.273 ( +5.8%) | 0.131 (−49.2%) |

Figure 2: **Concept descriptions better capture core features than default prompts.** The affinities between images and default prompts (`df`) are not stable w.r.t. changes in image background (BG) containing non-core features and are insensitive to changes in image foreground (FG) containing core features, as indicated by the relative changes (gray numbers in parentheses) w.r.t. the affinities of the original images. In contrast, the affinities between images and concept descriptions (`cd`) are stable w.r.t. to changes in BG while being highly responsive to changes in FG, making them a good detector for core features. See Appendix D.1 for more examples and a full quantitative study on this.

## 4 Fine-tuning zero-shot models with DRM

Zero-shot models like CLIP typically consist of an image encoder $f_\phi$ and a text encoder $g_\psi$ with parameters $\theta = (\phi, \psi)$. Image classification with such models is usually done by first creating a text prompt $t_y$ for each class label $y \in \mathcal{Y}$, and then assigning a probability for each $y$ to an image $x$ by

$$\hat{p}_\theta(y|x) = \frac{\exp(A_\theta(x, t_y)/\tau)}{\sum_{y' \in \mathcal{Y}} \exp(A_\theta(x, t_{y'})/\tau)}, \tag{5}$$

where $A_\theta(x, t_y) = \langle f_\phi(x), g_\psi(t_y) \rangle$ and $\tau$ is the temperature. The inner product $\langle f_\phi(x), g_\psi(t_y) \rangle$ can be intuitively understood as the *affinity* between $x$ and $t_y$, and we thus denote it as $A_\theta(x, t_y)$.

The classifier (5) was originally introduced by Radford et al. (2021) for zero-shot classification. We follow Goyal et al. (2023) to directly fine-tune this classifier, viewing the text embeddings $g_\psi(t_y)$ as the weights of a standard linear classification head for the image embeddings $f_\phi(x)$. Unlike standard classifiers, however, (5) additionally depends on the text prompt $t_y$ of which the design is important.

### 4.1 Dual prompts for fine-tuning zero-shot models

Let $\mathcal{T} = \{t_y \,|\, y \in \mathcal{Y}\}$ be the set of text prompts used to construct the classifier $p_\theta(y|x)$ according to Eq. (5). For such zero-shot classifiers, the general DRM objective (DRM) becomes

$$\min_{\theta \in \Theta} R_s(\theta; \mathcal{T}) + \lambda R_s^c(\theta; \mathcal{T}), \tag{6}$$

where the risks $R_s$ and $R_s^c$ not only depend on the model parameters $\theta$ but also on the prompts $\mathcal{T}$. For $R_s$, usually a set of *default prompts*, $\mathcal{T}^{\mathtt{df}} = \{t_y^{\mathtt{df}} \,|\, y \in \mathcal{Y}\}$, like "*an image of [class name]*" is used (Goyal et al., 2023; Oh et al., 2023). However, such prompts may not be suitable for $R_s^c$ as they are not specifically designed to bind with core features. In fact, as we will show, default prompts elicit representation that is biased towards non-core features. It is also know that non-visual and spurious descriptions contribute significantly to CLIP's representation (Esfandiarpoor et al., 2024).

To generate better prompts for $R_s^c$, we ask GPT-4 (Achiam et al., 2023) to describe the *core visual features* of each class, producing a set of *concept descriptions*, $\mathcal{T}^{\mathtt{cd}} = \{t_y^{\mathtt{cd}} \,|\, y \in \mathcal{Y}\}$. For example, to generate a concept description for *cougar*, we prompt GPT-4 with

> "*Q: Generate a short sentence that describes the visual features of Cougar. Do not include its function, its surroundings, or the environment it usually inhabits. The sentence should be concise. For example, [goldfish: a long, golden body with back fins].*"

and the concept description returned is

> "*a large, tawny cat with a muscular build and a small head.*" [3]

---

[3]More details about concept description generation can be found in Appendix C.

Figuratively, the text embedding $g_\psi(t_y^{\text{cd}})$ of the concept description represents the core features of the class from the text side. We use it to "pull out" the core features from the image embedding $f_\phi(x)$ via inner-product. As illustrated in Figure 2, the affinity between an image and its concept description is indeed a much better measure of the significance of core visual features. Regarding this point, a full quantitative study can be found in Appendix D.1.

Together, the two sets of prompts naturally give rise to the following objective:

$$\min_{\theta \in \Theta} R_s(\theta; \mathcal{T}^{\text{df}}) + \lambda R_s^c(\theta; \mathcal{T}^{\text{cd}}), \tag{7}$$

where we use default prompts $\mathcal{T}^{\text{df}}$ for ERM and concept descriptions $\mathcal{T}^{\text{cd}}$ for WRM. We do not use concept descriptions for ERM because it would erode the text-side core feature representation elicited by concept descriptions. This is supported by empirical evidence from our ablation study (Section 5.3) showing that (7) works best among various alternatives.

The dual prompts elicit separate predictions from the same model for the two sub-objectives of DRM. The ERM part, $R_s(\theta; \mathcal{T}^{\text{df}})$, is supervised by regular one-hot labels. The WRM part, $R_s^c(\theta; \mathcal{T}^{\text{cd}})$, is supervised by $p_c(y|x)$ as in (4). Next, we show how the same set of concept descriptions $\mathcal{T}^{\text{cd}}$ can be used to obtain a good estimate of $p_c(y|x)$.

## 4.2 Estimating $p_c(y|x)$ with concept descriptions

Recall that the oracle model $p_c(y|x)$ is based on a faithful representation of core features. Since we have demonstrated that concept descriptions bind well with core features on pre-trained CLIP models, a direct estimate for $p_c(y|x)$ can be obtained via (5) with $t_y \leftarrow t_y^{\text{cd}}$ and $\theta \leftarrow \theta_0$ where $\theta_0 = (\phi_0, \psi_0)$ denote the pre-trained CLIP parameters.

However, there is a crucial caveat. To illustrate, consider an image $x$ of class $y$. For another class $y'$ whose core features are *not* present in $x$, the affinity $A_{\theta_0}(x, t_{y'}^{\text{cd}})$ should ideally be very small. In practice, however, we find this is seldom the case. These extraneous affinity values, which we call *artifact terms*, often vary among classes and lead to poor estimates of $p_c(y|x)$ with high entropy.

To mitigate the impact of artifact terms, we perform a simple min-max normalization on $\xi(x, y) = \exp(A_{\theta_0}(x, t_y^{\text{cd}})/\tau)$ w.r.t. all training images $\mathcal{X}_y \subseteq \mathcal{X}$ labeled the same class $y$, as follows:

$$\gamma(x, y) = \frac{\xi(x, y) - \min_{x' \in \mathcal{X}_y} \xi(x', y)}{\max_{x' \in \mathcal{X}_y} \xi(x', y) - \min_{x' \in \mathcal{X}_y} \xi(x', y)}. \tag{8}$$

This effectively adjusts the affinity range of each class, reducing the difference in the artifact terms of different classes. Based on the normalization, the final estimation we propose for $p_c(y|x)$ is

$$\tilde{p}_c(y|x) = \begin{cases} \gamma(x, y), & y = y_x; \\ [1 - \gamma(x, y_x)] \cdot \frac{\gamma(x,y)}{\sum_{y' \neq y_x} \gamma(x, y')}, & y \neq y_x; \end{cases} \tag{9}$$

where $y_x$ is the ground-truth label of $x$. This ensures that for every class $y$, there exists at least one $x \in \mathcal{X}_y$ for which $\tilde{p}_c(y|x) = 1$, promoting balanced learning. When $\tilde{p}_c(y|x) < 1$ for $y = y_x$, the remaining probability is distributed to other classes $y \neq y_x$ according to the relative scale of the respective affinities.

We pre-compute the estimate $\tilde{p}_c(y|x)$ with the pre-trained CLIP model $\theta_0$ before fine-tuning, which is now the learning target of $R_s^c(\theta; \mathcal{T}^{\text{cd}})$. Since the computation requires the class labels, $\tilde{p}_c(y|x)$ can only be used for training (not for inference). Intuitively, learning from $\tilde{p}_c(y|x)$ can be seen as a form of self-distillation targeted at core features. In comparison, standard self-distillation methods (Furlanello et al., 2018; Zhang et al., 2019; Ji et al., 2021; Zhang et al., 2021) are largely indifferent to what specific information should be distilled.

## 4.3 Inference

The fine-tuning objective (7) involves two classifiers: the ERM classifier $\hat{p}_\theta^{\text{df}}(y|x)$ induced by $\mathcal{T}^{\text{df}}$, and the WRM classifier $\hat{p}_\theta^{\text{cd}}(y|x)$ induced by $\mathcal{T}^{\text{cd}}$. While either alone can be used for inference, we find that their mixture,

$$\hat{p}_\theta^{\text{dual}}(y|x) = \beta \cdot \hat{p}_\theta^{\text{df}}(y|x) + (1 - \beta) \cdot \hat{p}_\theta^{\text{cd}}(y|x), \tag{10}$$

where $\beta \in (0, 1)$, performs the best. This is expected as (10) essentially combines ERM with WRM as depicted in Figure 1. By default, we set $\beta = 1/(1 + \lambda)$ so to be as consistent with (7) as possible.

# 5 Experiments

In this section, we evaluate DRM on multiple real-world benchmarks and conduct ablation studies to assess the impacts of various design choices. We conduct our experiments on three varying sizes of pre-trained CLIP models: ViT-B/16, ViT-L/14 and ViT-L/14@336 (Radford et al., 2021). Finally, we analyze the reliability of LLM-generated concept descriptions and the impact of $\lambda$ on performance.

## 5.1 Setup

**Datasets.**    IMAGENET (Deng et al., 2009) comprises over a million natural images across 1,000 classes. We use the training set for fine-tuning and the validation set for assessing ID accuracy. For OOD evaluation, we consider ImageNet variants: IMAGENET-V2 (Recht et al., 2019), IMAGENET-R (Hendrycks et al., 2021a), IMAGENET-SKETCH (Wang et al., 2019), IMAGENET-A (Hendrycks et al., 2021b), and OBJECTNET (Barbu et al., 2019). We report accuracy for both ID/OOD performance.

WILDS-IWILDCAM (IWILDCAM) (Koh et al., 2021) contains camera-trap images for wildlife classification, with training images from 200 locations and OOD images from different locations. Both ID and OOD performances are measured using macro F1 scores.

WILDS-FMOW (FMOW) (Koh et al., 2021) is a dataset of satellite images from different years and continents for land use prediction. The dataset is split into training, validation, and testing domains based on the year of collection. There is also a notable shift between different continents. We report the ID testing accuracy and the worst-region OOD testing accuracy.

DOLLAR STREET-DA and GEOYFCC-DA (Prabhu et al., 2022) are datasets for testing model generalization from images in specific countries to new ones. For Dollar Street-DA, training images are from North America and Europe, with testing images from other continents. GeoYFCC-DA has a similar setup. Model effectiveness is measured by accuracy in seen and unseen countries.

**Baseline methods.**    The key baseline we compare our DRM method with is **FLYP** (Goyal et al., 2023). Following the FLYP paper, we include several baselines that do not utilize the text encoder. These methods are **LP** (linear probing), **FT** (fine-tuning), **L2-SP** (Li et al., 2018), and **LP-FT** (Kumar et al., 2022). In addition, we incorporate some more recent fine-tuning methods for zero-shot vision models. We also consider combining the weight-space averaging method, **WiSE-FT** (Wortsman et al., 2022), with DRM and the baselines. For more introduction to these methods, see Appendix B.

**Implementation details.**    We update both the image encoder and text encoder during fine-tuning, following FLYP (Goyal et al., 2023). Furthermore, FLYP uses the CLIP contrastive loss (Radford et al., 2021) instead of the standard cross-entropy loss. We adopt this approach for the ERM part of DRM to facilitate comparison. In short, when the hyperparameter $\lambda = 0$ in the DRM objective (7), the WRM loss term vanishes and DRM reduces to exactly FLYP.

We choose all hyperparameters of DRM and baseline methods based on the performance on the ID validation set, i.e., training-domain validation (Gulrajani and Lopez-Paz, 2021). The hyperparameter $\lambda$ of DRM is picked from $\{1, 2, 3, 4, 5\}$. More implementation details are presented in Appendix F.

## 5.2 Main results

We report the main results on IMAGENET, IWILDCAM, and FMOW in Table 1 and 2. The results on DOLLAR STREET-DA and GEOYFCC-DA can be found in Appendix E.2. We also compare DRM with some concurrent methods in Appendix E.3. All performance statistics, except some reported by previous papers (which we simply reuse), are averaged over 5 runs with different random seeds. The 95% confidence intervals over the 5 runs are reported.

Table 1 shows the results on CLIP ViT-B/16, the smallest of the three CLIP models. DRM achieves consistently better OOD performance than the baselines across all datasets, with and without WiSE-FT. Without WiSE-FT, DRM attains 5.0%, 12.4%, and 11.1% relative improvements over the best baseline method, FLYP, on the three benchmarks respectively. With WiSE-FT, the improvements remain significant at 1.9%, 11.6%, and 9.8% respectively. In terms of ID performance, DRM is roughly on par with FLYP, with a notable advantage on IWILDCAM.

Table 1: ID and OOD performances of DRM and baselines methods on CLIP ViT-B/16, with and without WiSE-FT. Best performances are highlighted in **bold**. For IMAGENET, we report the average performance over its 5 OOD test sets. Results on individual test sets are provided in Appendix E.1.

| | IMAGENET | | | | iWILDCAM | | | | FMoW | | | |
| | w/o WiSE-FT | | WiSE-FT | | w/o WiSE-FT | | WiSE-FT | | w/o WiSE-FT | | WiSE-FT | |
| Method | ID | OOD | ID | OOD | ID | OOD | ID | OOD | ID | OOD | ID | OOD |
|---|---|---|---|---|---|---|---|---|---|---|---|---|
| 0-shot | $68.3_{\pm0.0}$ | $58.7_{\pm0.0}$ | - | - | $8.7_{\pm0.0}$ | $11.0_{\pm0.0}$ | - | - | $20.4_{\pm0.0}$ | $18.7_{\pm0.0}$ | - | - |
| LP | $79.9_{\pm0.0}$ | $57.2_{\pm0.0}$ | $80.0_{\pm0.0}$ | $58.3_{\pm0.0}$ | $44.5_{\pm0.6}$ | $31.1_{\pm0.4}$ | $45.5_{\pm0.6}$ | $31.7_{\pm0.4}$ | $48.2_{\pm0.1}$ | $30.5_{\pm0.3}$ | $48.7_{\pm0.1}$ | $31.5_{\pm0.3}$ |
| FT | $81.4_{\pm0.1}$ | $54.8_{\pm0.1}$ | $82.5_{\pm0.1}$ | $61.3_{\pm0.1}$ | $48.1_{\pm0.5}$ | $35.0_{\pm0.5}$ | $48.1_{\pm0.5}$ | $35.0_{\pm0.5}$ | $68.5_{\pm0.1}$ | $39.2_{\pm0.7}$ | $68.5_{\pm0.1}$ | $41.5_{\pm0.5}$ |
| L2-SP | $81.6_{\pm0.1}$ | $57.9_{\pm0.1}$ | $82.2_{\pm0.1}$ | $58.9_{\pm0.1}$ | $48.6_{\pm0.4}$ | $35.3_{\pm0.3}$ | $48.6_{\pm0.4}$ | $35.3_{\pm0.3}$ | $68.6_{\pm0.1}$ | $39.4_{\pm0.6}$ | $68.4_{\pm0.1}$ | $40.3_{\pm0.6}$ |
| LP-FT | $81.8_{\pm0.1}$ | $60.5_{\pm0.1}$ | $82.1_{\pm0.1}$ | $61.8_{\pm0.1}$ | $49.7_{\pm0.5}$ | $34.7_{\pm0.4}$ | $50.2_{\pm0.5}$ | $35.7_{\pm0.4}$ | $68.4_{\pm0.2}$ | $40.4_{\pm1.0}$ | $68.5_{\pm0.2}$ | $42.4_{\pm0.7}$ |
| FLYP | $\mathbf{82.6}_{\pm0.0}$ | $60.2_{\pm0.1}$ | $82.9_{\pm0.0}$ | $63.2_{\pm0.1}$ | $52.2_{\pm0.6}$ | $35.6_{\pm1.2}$ | $52.5_{\pm0.6}$ | $37.1_{\pm1.2}$ | $68.6_{\pm0.2}$ | $41.3_{\pm0.8}$ | $\mathbf{68.9}_{\pm0.3}$ | $42.0_{\pm0.9}$ |
| DRM | $82.0_{\pm0.3}$ | $\mathbf{63.2}_{\pm0.2}$ | $82.4_{\pm0.2}$ | $\mathbf{64.0}_{\pm0.2}$ | $\mathbf{54.1}_{\pm0.5}$ | $\mathbf{40.0}_{\pm0.6}$ | $\mathbf{55.3}_{\pm0.4}$ | $\mathbf{41.4}_{\pm0.7}$ | $\mathbf{68.7}_{\pm0.3}$ | $\mathbf{45.9}_{\pm1.1}$ | $68.7_{\pm0.2}$ | $\mathbf{46.1}_{\pm0.8}$ |

Table 2: ID and OOD performances of DRM and FLYP on two larger CLIP models.

| Pre-trained model | Method | IMAGENET | | iWILDCAM | | FMoW | |
| | | ID | OOD | ID | OOD | ID | OOD |
|---|---|---|---|---|---|---|---|
| CLIP ViT-L/14 | FLYP | $84.6_{\pm0.3}$ | $73.4_{\pm0.1}$ | $56.0_{\pm1.1}$ | $41.9_{\pm0.7}$ | $71.2_{\pm0.5}$ | $48.2_{\pm0.5}$ |
| | FLYP+WiSE-FT | $85.1_{\pm0.2}$ | $75.1_{\pm0.1}$ | $57.2_{\pm0.7}$ | $42.1_{\pm0.5}$ | $\mathbf{72.0}_{\pm0.4}$ | $49.1_{\pm0.6}$ |
| | DRM | $85.0_{\pm0.2}$ | $75.5_{\pm0.2}$ | $\mathbf{61.8}_{\pm0.5}$ | $49.2_{\pm0.4}$ | $70.9_{\pm0.8}$ | $\mathbf{51.3}_{\pm0.7}$ |
| | DRM+WiSE-FT | $\mathbf{86.2}_{\pm0.1}$ | $\mathbf{76.2}_{\pm0.2}$ | $61.6_{\pm0.3}$ | $\mathbf{49.8}_{\pm0.4}$ | $71.4_{\pm0.5}$ | $\mathbf{51.3}_{\pm0.7}$ |
| CLIP ViT-L/14@336 | FLYP | $85.4_{\pm0.2}$ | $75.0_{\pm0.3}$ | $58.7_{\pm0.6}$ | $45.4_{\pm1.0}$ | $72.5_{\pm0.3}$ | $50.5_{\pm0.5}$ |
| | FLYP+WiSE-FT | $86.1_{\pm0.2}$ | $75.9_{\pm0.2}$ | $60.5_{\pm0.5}$ | $47.1_{\pm1.2}$ | $72.6_{\pm0.3}$ | $50.7_{\pm0.6}$ |
| | DRM | $85.9_{\pm0.1}$ | $76.0_{\pm0.2}$ | $\mathbf{62.8}_{\pm0.6}$ | $51.4_{\pm0.5}$ | $\mathbf{73.8}_{\pm0.5}$ | $52.5_{\pm0.9}$ |
| | DRM+WiSE-FT | $\mathbf{87.4}_{\pm0.0}$ | $\mathbf{77.1}_{\pm0.2}$ | $62.5_{\pm0.4}$ | $\mathbf{51.8}_{\pm0.5}$ | $\mathbf{73.8}_{\pm0.3}$ | $\mathbf{53.1}_{\pm0.6}$ |

Table 2 compares the performance of DRM and FLYP on two larger CLIP models. DRM again consistently outperforms FLYP in all cases. The previous state-of-the-art OOD performance for iWILDCAM and FMoW are 47.1 and 50.6 respectively, both achieved by FLYP+WiSE-FT with CLIP ViT-L/14@336. DRM improves those scores by 10.0% and 5.0% to 51.8 and 53.1 respectively.

Although DRM incurs more computational costs compared to FLYP due to an additional pass through the text encoder, the training and inference costs for DRM only increase by about 20% from FLYP (see Appendix F.3). This cost is insignificant relative to the performance gain.

Overall, our experiments suggest that DRM is highly effective in the robust fine-tuning of zero-shot models, outperforming previous methods by a large margin while maintaining scalability. As a bonus, we show in Appendix E.6 that DRM is also effective in fine-tuning ImageNet pre-trained CNNs.

### 5.3 Ablation study

We conduct our ablation study with CLIP ViT-L/14 on iWILDCAM. The main results are reported in Table 3 and discussed below. For additional results and details, see Appendix E.4 and E.5.

**Impact of dual risks.** The DRM objective (7) consists of an ERM term, $R_s(\theta; \mathcal{T}^{df})$, and a WRM term, $R_s^c(\theta; \mathcal{T}^{cd})$. From Table 3, we can see that, in terms of OOD performance, models fine-tuned with only the ERM term (Rows 4 & 5) significantly underperform models fine-tuned with both terms (Rows 1-3). Conversely, models fine-tuned with only the WRM term (Rows 6 & 7) have much worse ID performance. Although the OOD performance is improved, there is still a large gap from the DRM model (Row 1). Note that these hold regardless of the type of text prompts used by ERM/WRM.

**Impact of dual prompts for fine-tuning.** With both ERM and WRM in effect, we further investigate the impact of their prompts during fine-tuning. DRM uses two sets of prompts: default prompts $\mathcal{T}^{df}$ for ERM, and concept descriptions $\mathcal{T}^{cd}$ for WRM. Rows 8 & 9 of Table 3 show the performance of models fine-tuned using the same set of prompts for ERM and WRM. In either case, the model underperforms DRM (Row 1), validating the use of tailored prompts for specific learning targets.

Table 3: Results of ablation studies on DRM with CLIP ViT-L/14 performance and IWILDCAM. We use "df" and "cd" to denote the type of text prompts used to produce model predictions. "dual" refers to the mixture model (10) for inference. "–" means the corresponding loss term is not in use.

| Row | ERM $R_{\mathrm{s}}(\theta; \mathcal{T})$ | WRM $R_{\mathrm{s}}^{\mathrm{c}}(\theta; \mathcal{T})$ | Affinity norm. | Inference w/ model | Performance ID | OOD |
|---|---|---|---|---|---|---|
| 1 | | | | dual | **61.8** | **49.2** |
| 2 | df | cd | ✓ | df | 60.4 | 45.1 |
| 3 | | | | cd | 54.8 | 47.2 |
| 4 | df | – | – | df | 56.0 | 41.9 |
| 5 | cd | – | – | cd | 56.9 | 43.4 |
| 6 | – | df | ✓ | df | 52.4 | 45.3 |
| 7 | – | cd | ✓ | cd | 51.7 | 46.3 |
| 8 | df | df | ✓ | df | 54.4 | 45.1 |
| 9 | cd | cd | ✓ | cd | 54.0 | 46.0 |
| 10 | df | cd | ✗ | dual | 32.1 | 24.2 |

Intuitively, $\mathcal{T}^{\mathrm{cd}}$ is more aligned with WRM than $\mathcal{T}^{\mathrm{df}}$ as the learning targets (9) for WRM are based on the predictions of the pre-trained model prompted by $\mathcal{T}^{\mathrm{cd}}$. So, reducing the WRM term $R_{\mathrm{s}}^{\mathrm{c}}(\theta; \mathcal{T}^{\mathrm{cd}})$ in some sense limits the divergence of the predictions between the pre-trained models and the fine-tuned models, hence helping better preserve pre-trained (core) features. This explains why using $\mathcal{T}^{\mathrm{cd}}$ for both ERM and WRM would lead to poorer outcomes (Row 1 *vs*. 9). ERM targets, typically one-hot class labels, do not capture subtle core visual differences. As a result, ERM would weaken the bond between $\mathcal{T}^{\mathrm{cd}}$ and core visual features as perceived by the model during fine-tuning, and therefore reduce the power of WRM in preserving those features.

Interestingly, comparing Row 1 & 8, we find that $\mathcal{T}^{\mathrm{cd}}$ not only improves OOD performance but also ID performance. We hypothesize that this is because the core features are better preserved with $\mathcal{T}^{\mathrm{cd}}$ and thus the fine-tuned model relies on a more diverse set of features to make predictions, reducing overfitting and improving ID generalization as well.

**Impact of dual prompts for inference.** After DRM fine-tuning, we obtain a new CLIP model with updated parameters $\theta$. Since the model is fine-tuned with dual prompts, it is natural to use the same dual prompts for inference, as in (10). Indeed, we find that $\hat{p}_\theta^{\mathrm{dual}}$ (Row 1) outperforms $\hat{p}_\theta^{\mathrm{df}}$ (Row 2) and $\hat{p}_\theta^{\mathrm{cd}}$ (Row 3) in Table 3. In particular, while $\hat{p}_\theta^{\mathrm{df}}$ is better than $\hat{p}_\theta^{\mathrm{cd}}$ in-distribution and the latter is better out-of-distribution, they underperform $\hat{p}_\theta^{\mathrm{dual}}$ on both ID and OOD fronts. This is similar to the phenomenon we have observed in the fine-tuning scenario. Dual prompts generally reduce overfitting and the effect carries over from fine-tuning to inference.

**Impact of affinity normalization.** In Section 4.2, we pointed out a problematic issue regarding the direct estimate for $p_{\mathrm{c}}(y|x)$ obtained via (5) with $t_y \leftarrow t_y^{\mathrm{cd}}$ and $\theta \leftarrow \theta_0$. To address the issue, we proposed another estimate $\tilde{p}_{\mathrm{c}}(y|x)$ in (9) based on normalized affinities. Comparing the two approaches, Row 10 of Table 3 shows that the direct estimate leads to severe degradation in both ID and OOD performance, demonstrating the importance of affinity normalization.

### 5.4 Reliability of LLM-generated concept descriptions

**Consistency across repeated generations.** The concept descriptions used in our experiments are generated by GPT-4. Since the generation process is stochastic, it might impact the performance of DRM. To evaluate this impact, we repeatedly ask GPT-4 to generate a concept description for each IWILDCAM class for three times and then find the standard deviation of the resulting image-text affinities. The average standard deviation over 20,000 randomly sampled images of IWILDCAM is 0.0061, which is very small compared to the mean affinity, 0.2659. This suggests that the affinities are robust to the randomness in the generation process, which is therefore unlikely to have any noticeable

Table 4: Performance of fine-tuned CLIP ViT-L/14 on IWILDCAM with concept descriptions generated by different LLMs of various sizes.

| Method | LLM (#params) | ID | OOD |
|---|---|---|---|
| FLYP | N/A | 52.2 | 35.6 |
| DRM | GPT-3.5 (20B?) | 53.4 | 38.7 |
| DRM | GPT-4 (>1T?) | 54.1 | 40.0 |
| DRM | Llama-3 (8B) | 53.8 | 39.2 |
| DRM | Llama-3 (70B) | 54.0 | 39.9 |
| DRM | Llama-3 (405B) | 53.9 | 40.5 |

Table 5: Performance of DRM under different $\lambda$ on IWILDCAM and IMAGENET with CLIP ViT-L/14 and CLIP ViT-B/16 respectively.

| | IWILDCAM | | IMAGENET | |
|---|---|---|---|---|
| $\lambda$ | ID | OOD | ID | OOD |
| 0.0 | 56.0 | 41.9 | **82.6** | 60.2 |
| 1.0 | 59.1 | 47.3 | 81.5 | 62.5 |
| 2.0 | 60.0 | 48.1 | 81.8 | 63.1 |
| 3.0 | **61.8** | **49.2** | 82.0 | 63.2 |
| 4.0 | 60.9 | 48.6 | 81.9 | **63.4** |
| 5.0 | 60.1 | 48.5 | 81.7 | 63.3 |

impact on the performance of DRM. Examples of the generated descriptions in Appendix D.2 show that the same core visual features are described quite consistently across generations.

**Generation across different LLMs.** We also experiment with different LLMs of various sizes (from 8B to over 400B parameters) to generate concept descriptions. The results are reported in Table 4. While larger and more advanced models lead to better performance, DRM is not sensitive to the specific choice of LLM for generating the concept descriptions. Even with a relatively small LLM, Llama-3 (8B), DRM still maintains a significant edge over FLYP on IWILDCAM. Examples of the concept descriptions generated by the LLMs are provided in Appendix D.3.

## 5.5 Study on the effect of $\lambda$ in DRM

As in (7), DRM involves a hyperparameter $\lambda$ that balances the weight of the empirical risk and the worst-case risk. When $\lambda = 0$, only the empirical risk is involved during fine-tuning, resulting in an ERM model. As $\lambda$ increases, the influence of the empirical risk reduces, and the resulting model becomes closer to a WRM model. In practice, we choose the value of $\lambda > 0$ based on ID validation, which is often positively correlated with OOD performance (Taori et al., 2020; Miller et al., 2021). In Table 5, we show the ID and OOD performance of CLIP ViT-L/14 fine-tuned on IWILDCAM and CLIP ViT-B/16 fine-tuned on IMAGENET under different choices of $\lambda$.

Compared to FLYP ($\lambda = 0$), the results indicate that DRM maintains high-level OOD performance across $\lambda$ between 1 and 5, suggesting that DRM is fairly insensitive to the choice of $\lambda$. Furthermore, it is interesting to note that ID and OOD performance improve together on IWILDCAM as $\lambda$ increases to 3.0. Our further analysis reveals that this occurs because DRM assists in reducing overfitting—there is a decrease in training accuracy (from 88.29 to 87.41) and an increase in validation accuracy (from 81.64 to 82.43) as $\lambda$ increases (from 1 to 3)—thereby enhancing both ID and OOD performance. While DRM does not always improve ID performance (e.g., on IMAGENET), the result suggests that it still achieves a good trade-off between ID and OOD performance (-0.6 ID performance for +3.2 OOD performance), which in turn could lead to a better Pareto front.

## 6 Conclusion

In conclusion, this paper introduces dual risk minimization (DRM), a novel method that enhances the robustness of models fine-tuned from zero-shot foundation models against distribution shifts. DRM combines ERM with WRM, focusing on the preservation of core features which essentially define the target classes. To guide the fine-tuning process, DRM utilizes concept descriptions generated by LLMs like GPT-4. By balancing expected and worst-case performance, DRM overcomes the traditional limitations of ERM and achieves significant OOD performance improvements on multiple real-world benchmarks, establishing a new state of the art. Potential future directions include a deeper theoretical investigation into DRM as a general principle, using DRM to improve the general robustness of zero-shot models across a broad range of tasks, and better understanding the role of concept descriptions in vision-language modeling.

## Acknowledgements

This research was supported in part by Hong Kong Research Grants Council under grant 16204920. Kaican Li and Weiyan Xie were supported in part by the Huawei PhD Fellowship Scheme. Ricardo Silva acknowledges support of the UKRI AI programme, and the Engineering and Physical Sciences Research Council, for CHAI - EPSRC AI Hub for Causality in Healthcare AI with Real Data (grant number EP/Y028856/1).

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

# A  Proofs

**Lemma 1.** *Let $p$ and $q$ be two probability distributions over $\mathcal{X} \times \mathcal{Y}$. The cross-entropy between $p(y|x)$ and $q(y|x)$ over $p(x)$, i.e., $H_p(q) = \mathbb{E}_{(x,y)\sim p}[-\log q(y|x)]$, is convex w.r.t. $q$.*

*Proof.* It suffices to show that for any pair of $(q_1, q_2)$ and $\alpha \in [0, 1]$ we have $H_p(\alpha q_1 + (1-\alpha)q_2) \leq \alpha H_p(q_1) + (1-\alpha)H_p(q_2)$.

$$
\begin{aligned}
H_p(\alpha q_1 + (1-\alpha)q_2) &= \mathbb{E}_{(x,y)\sim p}[-\log(\alpha q_1(y|x) + (1-\alpha)q_2(y|x))] \\
&\leq \mathbb{E}_{(x,y)\sim p}[-\alpha \log q_1(y|x) - (1-\alpha)\log q_2(y|x)] \quad (11) \\
&= \alpha H_p(q_1) + (1-\alpha)H_p(q_2). \qquad \square
\end{aligned}
$$

**Theorem 1.** *Strong duality holds between IDRM and the following dual problem:*

$$
\max_{\lambda' \geq 0} \min_{\theta \in \Theta} \left[ R_{\mathrm{s}}(\theta) + \lambda' \max_{d \in \mathcal{D}} R_d(\theta) \right] - \lambda'\alpha. \tag{3}
$$

*Proof.* Recall that IDRM aims to solve for

$$
\min_{\theta \in \Theta} R_{\mathrm{s}}(\theta) \quad \text{subject to} \quad \max_{d \in \mathcal{D}} R_d(\theta) \leq \alpha. \tag{IDRM}
$$

Here, $R_d(\theta) = H_{p_d}(\hat{p}_\theta(y|x)) = \mathbb{E}_{(x,y)\sim p_d}[-\log \hat{p}_\theta(y|x)]$ is the cross-entropy between $p_d(y|x)$ and $\hat{p}_\theta(y|x)$ over $p_d(x)$. It follows from Lemma 1 that $R_d(\theta)$ is convex w.r.t. $\hat{p}_\theta(y|x)$ for all $d \in \mathcal{D}$.

Since the point-wise maximum of multiple convex functions is also convex, $\max_{d \in \mathcal{D}} R_d(\theta)$ is convex and therefore IDRM is a convex optimization problem w.r.t. $\hat{p}_\theta(y|x)$. By Slater's condition, strong duality holds between IDRM and the Lagrangian dual of IDRM:

$$
\max_{\lambda' \geq 0} \min_{\theta \in \Theta} R_{\mathrm{s}}(\theta) + \lambda' \left[ \max_{d \in \mathcal{D}} R_d(\theta) - \alpha \right], \tag{12}
$$

for any $\alpha > \min_{\theta \in \Theta} \max_{d \in \mathcal{D}} R_d(\theta)$, i.e., when a strictly feasible solution to (IDRM) exists. $\qquad \square$

# B  Introduction to baseline methods

The key baseline we compare our DRM method with is "fine-tune like you pre-train" (FLYP) (Goyal et al., 2023). While traditional fine-tuning of CLIP models adds a randomly initialized classification head on top of the image encoder, Goyal et al. (2023) demonstrated that it is more effective to simply reuse the text encoder. We therefore follow FLYP to update both the image encoder and text encoder in fine-tuning. They now differ only in the loss functions used in fine-tuning. FLYP uses the CLIP contrastive loss (Radford et al., 2021) for the ERM and showed that this is better than using the standard cross-entropy loss. We adopt this approach for the ERM component (the first term) of DRM to facilitate comparison. Thus, the only difference between DRM and FLYP is the additional regularization term in the DRM loss function.

Before the introduction of FLYP, conventional fine-tuning of CLIP models involved adding a linear classification head to the image encoder. The linear probing method (LP) only fine-tunes this new classification head, keeping the image encoder fixed, whereas the full fine-tuning method (FT) trains both the head and the encoder. The LP-FT approach (Kumar et al., 2022) begins with LP and then transitions to full fine-tuning.

Besides, L2-SP (Li et al., 2018) and WiSE-FT (Wortsman et al., 2022) are two established fine-tuning variants that restrict the divergence from the pre-trained model. L2-SP specifically integrates an $L^2$ regularization term into the loss function to constrain the parameter shifts of the fine-tuned model relative to the pre-trained model. WiSE-FT (Wortsman et al., 2022) interpolates parameters of a pre-trained zero-shot model $\theta_{\mathrm{zs}}$ and that of a fine-tuned model $\theta_{\mathrm{ft}}$ using $\theta_{\mathrm{wise\text{-}ft}} = \rho \cdot \theta_{\mathrm{zs}} + (1-\rho) \cdot \theta_{\mathrm{ft}}$.

Alongside L2-SP and WiSE-FT, several works concurrent with our own have also introduced robust fine-tuning methods via reducing the difference between pre-trained and fine-tuned models. For example, CAR-FT (Mao et al., 2024) and the method proposed by Cheng et al. (2024) seek to minimize the distance between the context distributions generated by pre-trained and fine-tuned CLIP

models. However, a significant limitation of them is that they require prior knowledge of image contexts, such as background and viewpoint, which restricts their practical applicability.

Alternatively, Lipsum-ft (Nam et al., 2024) implements a regularization strategy without requiring prior context information, focusing on minimizing the $L^2$ distance between the affinities of images and random texts from both pre-trained and fine-tuned models. CLIPood (Shu et al., 2023) utilizes a beta moving average for updating parameters during training, and CaRot (Oh et al., 2023) focuses on regularizing singular value distributions while incorporating an exponential moving average for parameter updates. TPGM (Tian et al., 2023a) and FTP (Tian et al., 2023b) take a more fine-grained approach by autonomously determining the most effective regularization for each layer's parameters and enabling more efficient learning of layer-specific projection regularization, respectively.

While these methods aim to prevent the fine-tuned model from deviating too far from the pre-trained one, they do not specify which pre-trained features should be preserved. In contrast, our proposed DRM focuses on preserving the dataset-related core visual features, making it a more targeted approach compared to the aforementioned methods.

## C   Details on concept description creation

We utilized GPT-4 (Achiam et al., 2023), a leading-edge language model developed by OpenAI, to create concept descriptions. We supplied GPT-4 with the class name and tailored prompts designed to guide the model to focus on the essential visual features of the class while omitting unrelated contextual information.

Specifically, for the iWildCam dataset, our prompt to GPT-4 was:

> Q: Generate a short sentence that describes the visual features of a given animal. Do not include its function, its surroundings, or the environment it usually inhabits. The sentence should be concise. For example, [goldfish: a long, golden body with back fins].

After that, we asked it to generate concept description for each wildlife class. For example:

> Q: Now the given animal is *Cougar*.
> A: [*Cougar*: *a large, tawny cat with a muscular build and a small head.*]

Another example is the FMoW dataset. We found it is challenging to encapsulate the core visual features of land used for specific purposes in just one short sentence. Consequently, we employed GPT-4 to generate six sentences for each type of land use. When we implemented DRM on this dataset, we slightly modified the image classifier to use the average of the six text embeddings for each class as the classification head.

For this dataset, our initial prompt to GPT-4 was:

> Q: The Functional Map of the World (fMoW) dataset is designed to identify the functions of buildings and land use by examining satellite images over time.
>
> Your task is to generate six sentences, each describing the visual features likely to be seen in these satellite images, given the functional purpose of buildings and land use. The features you describe should be visible from an aerial perspective and consistent across various countries. Each sentence should clearly and comprehensively describe a visual feature, focusing solely on its appearance without including details of the surrounding area or the purpose of the features, such as "Helipad for urgent aerial access".

After that, we asked it to generate concept description for each land use. For example:

> Q: Now the given land use is *Road Bridge*.
> A: [*Road Bridge*:
> *Visibly characterized by a straight or slightly curved linear structure that spans across natural or man-made obstacles.*

*Constructed from materials that appear robust in satellite images, often showing as grey or dark tones against the landscape.*

*The bridge deck is typically marked with parallel lines, indicating the lanes for vehicular traffic visible from above.*

*Elevation above the surrounding area is noticeable, with shadows often highlighting the height and structure of the bridge.*

*Support features like piers or columns are regularly spaced, visible as distinct vertical elements that support the span.*

*End points of the bridge integrate with road networks, appearing as seamless transitions from elevated to ground-level roads.*]

The full list of concept descriptions generated by GPT-4 for each class of different datasets considered in this work will be publicly available online later.

By utilizing GPT-4, we were able to automatically produce precise concept descriptions of various classes. The use of LLMs to generate class description from the class name is not new, which have been explored in Menon and Vondrick (2022); Pratt et al. (2023); Maniparambil et al. (2023) and Yang et al. (2023); Yan et al. (2023). Among them, the first three works mainly explored the use of class descriptions for zero-shot classification, while this work focuses on fine-tuning. The class descriptions generated in these studies often include contextual information due to prompts like "*Describe an image from the internet of a(n) ...*", which may not be suitable for our application of estimating worst-case risk. The descriptions in Yang et al. (2023); Yan et al. (2023) are more aligned with our approach, while they focus on using LLM-generated concept descriptions to develop concept bottleneck models for interpretable image classification.

## D    Empirical study on concept descriptions

### D.1    Qualitative and quantitative study on the reliability of concept descriptions

Complementing Figure 2, Figure 3 provides additional examples indicating that with the pre-trained CLIP models, concept descriptions have the ability to extract core features. To further support this claim, we present a full quantitative study. The study is conducted with Hard ImageNet (Moayeri et al., 2022) and consists of two parts. First, we remove image background (BG), and observing how the image-text affinities change for default prompts (df) and concept descriptions (cd) respectively. In the second part, we do the same but with foreground (FG) removed.

Table 6: Quantitative study on the reliability of concept descriptions verse default prompts. The affinities to concept descriptions are sensitive to changes in the core features of image foregrounds (FG) and remain relatively stable against changes in the non-core features of backgrounds (BG).

|     | FG & BG | w/o BG | w/o FG |
| --- | --- | --- | --- |
| df | 0.3473 | 0.2393 (-31.1%) | 0.3407 (-1.9%) |
| cd | 0.2660 | 0.2387 (-10.3%) | 0.1180 (-55.6%) |

Table 6 shows the average affinities over all 19,097 images across all 15 classes of Hard ImageNet. The percentages in the table indicate the relative changes w.r.t. the affinities of the original images (FG & BG). The result shows that the affinities of concept descriptions are much more invariant to changes in non-core features than default prompts (-10.3% vs. -31.1%). Moreover, the affinities of concept descriptions are quite responsive (-55.6%) to changes in core features. In contrast, the affinities of default prompts barely change (-1.9%) in response to the absence of core features. These results indicate that the affinities associated with concept descriptions are reliable indicators of core visual features in the images. In contrast, the affinities from default prompts are more sensitive to the changes in non-core visual features.

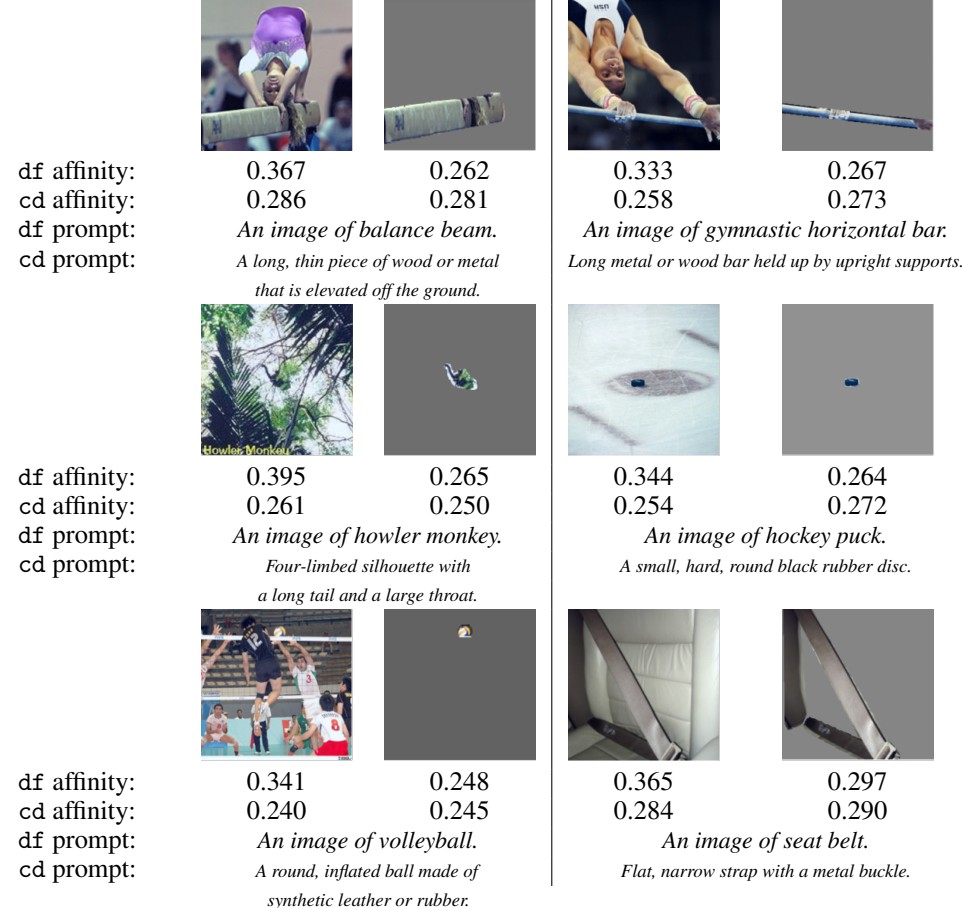

| | | | | |
|---|---|---|---|---|
| df affinity: | 0.367 | 0.262 | 0.333 | 0.267 |
| cd affinity: | 0.286 | 0.281 | 0.258 | 0.273 |

df prompt: *An image of balance beam.*  *An image of gymnastic horizontal bar.*

cd prompt: *A long, thin piece of wood or metal*  *Long metal or wood bar held up by upright supports.*
*that is elevated off the ground.*

| | | | | |
|---|---|---|---|---|
| df affinity: | 0.395 | 0.265 | 0.344 | 0.264 |
| cd affinity: | 0.261 | 0.250 | 0.254 | 0.272 |

df prompt: *An image of howler monkey.*  *An image of hockey puck.*

cd prompt: *Four-limbed silhouette with*  *A small, hard, round black rubber disc.*
*a long tail and a large throat.*

| | | | | |
|---|---|---|---|---|
| df affinity: | 0.341 | 0.248 | 0.365 | 0.297 |
| cd affinity: | 0.240 | 0.245 | 0.284 | 0.290 |

df prompt: *An image of volleyball.*  *An image of seat belt.*

cd prompt: *A round, inflated ball made of*  *Flat, narrow strap with a metal buckle.*
*synthetic leather or rubber.*

Figure 3: Concept description prompts (cd) yield affinities which are more robust to the change of context information than the affinities yielded by the default text prompts (df).

## D.2  Examples of repeated generations of concept descriptions

To evaluate the stochasticity of LLM in generating concept descriptions, we repeatedly ask GPT-4 to generate concept descriptions for each iWildCam class. Below are some examples:

*White-lipped Peccary:*

- **Output 1:** *Compact, dark grey body with distinctive white markings around the mouth.*
- **Output 2:** *Compact, dark gray body with distinctive white markings around the lips.*
- **Output 3:** *A stocky body with coarse, dark hair and distinct white markings around the mouth.*

*Waterbuck:*

- **Output 1:** *Stocky body with long fur, a white ring on the rump, and shaggy brown coat.*
- **Output 2:** *Thick, shaggy brown coat with a white ring around the rump and long horns.*
- **Output 3:** *A robust antelope with a shaggy brown coat and a white ring around the rump.*

These examples show that while GPT-4's outputs for the same class vary slightly, they consistently highlight key visual features. The variations are primarily in the language details, such as the order of features or the terminology used. For example, for the waterbuck, the consistent visual features across all three outputs are the "stocky (thick, robust) body", "white ring on (around) the rump", and "shaggy brown coat".

### D.3 Examples to generate concept descriptions with different LLMs

The examples and experiment results reported in our paper are based on concept descriptions generated by GPT-4. We now consider different LLMs to generate concept descriptions for the classes in the iWildCam dataset. We present some examples and our findings below.

**Examples:**

*White-lipped Peccary:*

- **GPT-4:** *Compact, dark grey body with distinctive white markings around the mouth.*
- **GPT-3.5-Turbo:** *Grayish body with a white stripe across the lips.*
- **Llama-3-8b:** *Reddish-brown fur with a distinctive white stripe.*
- **Llama-3-70b:** *Dark brown pig-like animal with a distinctive white stripe on the lips.*
- **Llama-3-405b:** *Stout, brown body with a distinctive white stripe on the lips and sharp tusks.*

*Waterbuck:*

- **GPT-4:** *Stocky body with long fur, a white ring on the rump, and shaggy brown coat.*
- **GPT-3.5-Turbo:** *Antelope species with a shaggy coat and a white ring on its rump.*
- **Llama-3-8b:** *Reddish-brown coat with a distinctive horn shape and a long, slender body.*
- **Llama-3-70b:** *Large, brown antelope with a distinctive white ring on its rump and a long, shaggy coat.*
- **Llama-3-405b:** *Large, brown body with a distinctive pair of curved horns and a white ring on the rump.*

*Zebra:*

- **GPT-4:** *Distinctive black and white striped coat with an upright mane and slender legs.*
- **GPT-3.5-Turbo:** *Medium-sized equid with black and white stripes.*
- **Llama-3-8b:** *Black and white striped coat with a distinctive mane and a long, slender body.*
- **Llama-3-70b:** *Black and white striped equine with a distinctive body shape and a long mane.*
- **Llama-3-405b:** *Medium-sized, black and white body with a distinctive striped pattern and a long, flowing mane.*

**Findings:**

1. The concept descriptions generated by different LLMs capture common key visual features for the same class, such as "white stripe" for white-lipped peccary, "white ring on the rump" for waterbuck, and "black and white stripes" for zebra.

2. Comparing outputs of GPT-3.5-Turbo and GPT-4, the outputs generated by GPT-4 cover more visual details, while those by GPT-3.5-Turbo are generally shorter.

3. As model size increases, from Llama-3-8b to Llama-3-405b, the generated outputs become more detailed. For instance, Llama-3-8b mentions "white stripe" but does not specify its location on the lips as Llama-3-70b/405b do. Additionally, Llama-3-8b sometimes makes factual errors, such as describing the incorrect color for white-lipped peccary and waterbuck.

4. GPT-3.5-Turbo sometimes deviated from the prompt instructions, generating outputs that include non-visual features, such as the "distinctive call" of the *Great Tinamou*, which are not visible.

Despite variations in the quality of concept descriptions generated by different LLMs, as discussed in Section 5.4, DRM-trained models using descriptions from any of these tested LLMs consistently show significant improvements in OOD performance compared to FLYP-trained models. This demonstrates that the effectiveness of DRM-trained models is not sensitive to the choice of the LLM.

# E  Additional experiment results

## E.1  Detailed performance on ImageNet OOD test sets

The average accuracy across the five ImageNet OOD test sets has been presented in Table 1. We report the detailed results for each OOD test set in Table 7. Without WiSE-FT, DRM substantially outperforms the previous best fine-tuning results by FLYP on ImageNet-R and ImageNet-A, with increases from 71.4 to 77.8 and from 48.1 to 53.3, respectively. Meanwhile, the ID performance is at a comparable level. With WiSE-FT, the improvements remain significant, rising from 76.0 to 79.5 on ImageNet-R and from 53.0 to 54.2 on ImageNet-A.

Table 7: Performance on ImageNet OOD variants with CLIP ViT-B/16. "OOD" stands for the average performance over the OOD datasets.

| Method | w/o WiSE-FT | | | | | | | WiSE-FT | | | | | | |
| | ID | Im-V2 | Im-R | Im-A | Sketch | ONet | OOD | ID | Im-V2 | Im-R | Im-A | Sketch | ONet | OOD |
|---|---|---|---|---|---|---|---|---|---|---|---|---|---|---|
| 0-shot | 68.3 | 61.9 | 77.7 | 50.0 | 48.3 | 55.4 | 58.7 | 68.3 | 61.9 | 77.7 | 50.0 | 48.3 | 55.4 | 58.7 |
| LP | 79.9 | 69.8 | 70.8 | 46.4 | 46.9 | 52.1 | 57.2 | 80.0 | 70.3 | 72.4 | 47.8 | 48.1 | 52.8 | 58.3 |
| FT | 81.3 | 71.2 | 66.1 | 37.8 | 46.1 | 53.3 | 54.9 | 82.5 | 72.8 | 74.9 | 48.1 | 51.9 | 59.0 | 61.3 |
| L2-SP | 81.7 | 71.8 | 70.0 | 42.5 | 48.5 | 56.2 | 57.8 | 82.2 | 72.9 | 75.1 | 48.6 | 51.4 | 58.9 | 61.4 |
| LP-FT | 81.7 | 72.1 | 73.5 | 47.6 | 50.3 | 58.2 | 60.3 | 82.1 | 72.8 | 75.3 | 50.1 | 51.7 | 59.2 | 61.8 |
| FLYP | **82.6** | 73.0 | 71.4 | 48.1 | 49.6 | **58.7** | 60.2 | **82.9** | 73.5 | 76.0 | 53.0 | 52.3 | **60.8** | 63.1 |
| DRM | 82.0 | **73.4** | **77.8** | **53.3** | **52.5** | 58.6 | **63.2** | 82.4 | **73.9** | **79.5** | **54.2** | 52.8 | 59.7 | **64.0** |

## E.2  Performance on Dollar Street-DA and GeoYFCC-DA

We followed the train-test split outlined by Prabhu et al. (2022). As there was no dedicated validation set, we split 20% of the training set for validation purposes. The ID and OOD performance results are reported based on the ID performance on the validation set and the OOD performance on the test set, which consists of images from countries not included in the training and validation sets.

The results presented in Table 8 demonstrate that, compared to FLYP-trained models, DRM-trained models exhibit improved performance on images from new countries.

Table 8: ID and OOD performance on Dollar Street-DA and GeoYFCC-DA with CLIP ViT-B/16.

| Method | Dollar Street-DA | | GeoYFCC-DA | |
| | ID | OOD | ID | OOD |
|---|---|---|---|---|
| 0-shot | $64.0_{\pm 0.0}$ | $53.7_{\pm 0.0}$ | $56.2_{\pm 0.0}$ | $52.3_{\pm 0.0}$ |
| FLYP | $\mathbf{82.4}_{\pm \mathbf{0.3}}$ | $71.8_{\pm 0.2}$ | $71.0_{\pm 0.3}$ | $58.0_{\pm 0.3}$ |
| FLYP+WiSE-FT | $\mathbf{82.4}_{\pm \mathbf{0.2}}$ | $72.7_{\pm 0.2}$ | $71.2_{\pm 0.3}$ | $58.7_{\pm 0.2}$ |
| DRM | $81.4_{\pm 0.2}$ | $73.9_{\pm 0.3}$ | $\mathbf{71.8}_{\pm \mathbf{0.4}}$ | $62.5_{\pm 0.4}$ |
| DRM+WiSE-FT | $82.0_{\pm 0.1}$ | $\mathbf{74.7}_{\pm \mathbf{0.2}}$ | $\mathbf{71.8}_{\pm \mathbf{0.3}}$ | $\mathbf{63.0}_{\pm \mathbf{0.2}}$ |

## E.3  Comparison to some more recent methods

As discussed in Appendix B, there are some more recent robust fine-tuning methods. We include a comparison to some of those methods based on the results of fine-tuning CLIP ViT-B/16 on iWildCam and FMoW datasets. The results are reported in Table 9. The results clearly show that, the more recent methods still significantly lag behind DRM in term of OOD performance.

Table 9: Performance results for iWildCam and FMoW with CLIP ViT-B/16 including some more recent methods.

| Method | iWildCam | | | | FMoW | | | |
| | w/o WiSE-FT | | WiSE-FT | | w/o WiSE-FT | | WiSE-FT | |
| | ID | OOD | ID | OOD | ID | OOD | ID | OOD |
|---|---|---|---|---|---|---|---|---|
| 0-shot | $8.7_{\pm0.0}$ | $11.0_{\pm0.0}$ | - | - | $20.4_{\pm0.0}$ | $18.7_{\pm0.0}$ | - | - |
| LP | $44.5_{\pm0.6}$ | $31.1_{\pm0.4}$ | $45.5_{\pm0.6}$ | $31.7_{\pm0.4}$ | $48.2_{\pm0.1}$ | $30.5_{\pm0.3}$ | $48.7_{\pm0.1}$ | $31.5_{\pm0.3}$ |
| FT | $48.1_{\pm0.5}$ | $35.0_{\pm0.5}$ | $48.1_{\pm0.5}$ | $35.0_{\pm0.5}$ | $68.5_{\pm0.1}$ | $39.2_{\pm0.7}$ | $68.5_{\pm0.1}$ | $41.5_{\pm0.5}$ |
| L2-SP | $48.6_{\pm0.4}$ | $35.3_{\pm0.3}$ | $48.6_{\pm0.4}$ | $35.3_{\pm0.3}$ | $68.6_{\pm0.1}$ | $39.4_{\pm0.6}$ | $68.4_{\pm0.1}$ | $40.3_{\pm0.6}$ |
| LP-FT | $49.7_{\pm0.5}$ | $34.7_{\pm0.4}$ | $50.2_{\pm0.5}$ | $35.7_{\pm0.4}$ | $68.4_{\pm0.2}$ | $40.4_{\pm1.0}$ | $68.5_{\pm0.2}$ | $42.4_{\pm0.7}$ |
| FLYP | $52.2_{\pm0.6}$ | $35.6_{\pm1.2}$ | $52.5_{\pm0.6}$ | $37.1_{\pm1.2}$ | $68.6_{\pm0.2}$ | $41.3_{\pm0.8}$ | $68.9_{\pm0.3}$ | $42.0_{\pm0.9}$ |
| CLIPood | $48.4_{\pm0.4}$ | $36.1_{\pm0.4}$ | $48.3_{\pm0.3}$ | $36.5_{\pm0.4}$ | $68.2_{\pm0.3}$ | $40.8_{\pm0.9}$ | $68.3_{\pm0.3}$ | $41.2_{\pm0.7}$ |
| TPGM | $47.5_{\pm0.3}$ | $35.9_{\pm0.4}$ | $46.8_{\pm0.3}$ | $36.2_{\pm0.3}$ | $68.4_{\pm0.3}$ | $39.6_{\pm0.8}$ | $67.8_{\pm0.2}$ | $39.9_{\pm0.7}$ |
| LipSum-FT | $50.7_{\pm0.8}$ | $36.6_{\pm0.7}$ | $48.4_{\pm0.5}$ | $36.9_{\pm0.6}$ | $68.4_{\pm0.3}$ | $41.3_{\pm1.0}$ | $68.1_{\pm0.3}$ | $42.0_{\pm0.5}$ |
| CaRot | $49.7_{\pm0.4}$ | $34.3_{\pm0.3}$ | $48.3_{\pm0.3}$ | $34.7_{\pm0.3}$ | $68.8_{\pm0.2}$ | $39.8_{\pm0.6}$ | $68.3_{\pm0.2}$ | $40.7_{\pm0.5}$ |
| DRM | $\mathbf{54.1_{\pm0.5}}$ | $\mathbf{40.0_{\pm0.6}}$ | $\mathbf{55.3_{\pm0.4}}$ | $\mathbf{41.4_{\pm0.7}}$ | $\mathbf{68.7_{\pm0.3}}$ | $\mathbf{45.9_{\pm1.1}}$ | $68.7_{\pm0.2}$ | $\mathbf{46.1_{\pm0.8}}$ |

## E.4 Full Ablation Study

**Setup.** Given a labeled dataset $\{(x_i, y_i)\}_{i=1}^N$ sampled from the training domain $d_s$, the final DRM objective (7), i.e., $R_s(\theta; \mathcal{T}^{\mathrm{df}}) + \lambda R_s^c(\theta; \mathcal{T}^{\mathrm{cd}})$, for fine-tuning zero-shot models can be expanded as

$$\frac{1}{N}\sum_{i=1}^N\Big[-\log\hat{p}_\theta^{\mathrm{df}}(y_i|x_i) - \lambda\sum_{y'\in\mathcal{Y}}\hat{p}_{\theta_0}^{\mathrm{pr}}(y'|x_i)\log\hat{p}_\theta^{\mathrm{cd}}(y'|x_i)\Big], \qquad (13)$$

where $\hat{p}_\theta^{\mathrm{df}}(y|x)$ and $\hat{p}_\theta^{\mathrm{cd}}(y|x)$ are the classifiers (5) induced by the default prompts $\mathcal{T}^{\mathrm{df}} = \{t_y^{\mathrm{df}} \mid y \in \mathcal{Y}\}$ and the concept descriptions $\mathcal{T}^{\mathrm{cd}} = \{t_y^{\mathrm{cd}} \mid y \in \mathcal{Y}\}$, respectively; and $\hat{p}_{\theta_0}^{\mathrm{pr}}(y|x) = \tilde{p}_c(y|x)$ which is defined by (9) to estimate $p_c(y|x)$. Here, we use $\hat{p}_{\theta_0}^{\mathrm{pr}}(y|x)$ (where pr stands for 'proxy') instead of $\tilde{p}_c(y|x)$ to ease the discussion of possible variations of DRM.

Consider the following generalized form of (13) with three varying options, t1 and t2 indicating the classifier types defined with different sets of text prompts, and type indicating the type of model used as the proxy for $p_c(y|x)$:

$$\frac{1}{N}\sum_{i=1}^N\Big[-\log\hat{p}_\theta^{\mathrm{t1}}(y_i|x_i) - \lambda\sum_{y'\in\mathcal{Y}}\hat{p}_{\theta_0}^{\mathrm{type}}(y'|x_i)\log\hat{p}_\theta^{\mathrm{t2}}(y'|x_i)\Big], \qquad (14)$$

As stated in (13), our final DRM training objective (7) uses t1 = df in the ERM term, with t2 = cd and type = pr in the regularization term. We denote this as our standard setting, (S) in short. We conduct the following ablation study with the pre-trained CLIP ViT-L/14 and fine-tune the model on the iWildCam dataset, with results presented in Table 10.

**(a) Inference options after dual classifier training:** Two classifiers are involved in our DRM training: $\hat{p}_\theta^{\mathrm{df}}(y|x)$ and $\hat{p}_\theta^{\mathrm{cd}}(y|x)$. As outlined in (10), we combine both classifiers for inference. An alternative is to only use one of the two classifiers for inference. We denote inference with only $\hat{p}_\theta^{\mathrm{df}}(y|x)$ as (a1), and with only $\hat{p}_\theta^{\mathrm{cd}}(y|x)$ as (a2). The comparison between (a1), (a2), and (S) in Table 10 shows combining both classifiers for inference enhances both ID and OOD performance compared to using either alone. This reveals that the two classifiers have a complementary effect as illustrated in Figure 1, and corroborates our view that ERM and WRM are both vital to OOD robustness.

**(b) Vanilla DRM using a single set of text prompts:** In our standard DRM setting (S), t1 = df and t2 = cd. The vanilla DRM we discussed in Section 4 uses t1 = t2 = df. Alternatively, one can also consider t1 = t2 = cd. We experiment with these two alternative settings denoted by (b1) and (b2) in Table 10. The contrast between (b1) and (S) confirms our intuition: using the concept

Table 10: Ablation study on DRM with CLIP ViT-L/14 (w/o WiSE-FT) on iWildCam.

| General setting | | | Specification | | | | | Performance | |
|---|---|---|---|---|---|---|---|---|---|
| | | | t1 | t2 | type | Classifier comb. | Infer w/ | ID | OOD |
| Standard DRM | | (S) | df | cd | pr | joint training | (10) | **61.8** | **49.2** |
| (a) | Infer with one classifier after dual classifier training | (a1) | df | cd | pr | joint training | df | 60.4 | 45.1 |
| | | (a2) | df | cd | pr | joint training | cd | 54.8 | 47.2 |
| (b) | Vanilla DRM using one set of text prompts | (b1) | df | df | pr | joint training | df | 54.4 | 45.1 |
| | | (b2) | cd | cd | pr | joint training | cd | 54.0 | 46.1 |
| (c) | Use different proxy models | (c1) | df | cd | cd | joint training | (10) | 32.1 | 24.2 |
| | | (c2) | df | cd | pr-df | joint training | (10) | 54.4 | 45.1 |
| | | (c3) | df | cd | one-hot | joint training | (10) | 57.3 | 45.1 |
| (d) | Use only one risk for training | (d1) | df | / | / | / | df | 56.0 | 41.9 |
| | | (d2) | cd | / | / | / | cd | 56.9 | 43.4 |
| | | (d3) | / | cd | pr | / | cd | 51.7 | 46.3 |
| (e) | Combine independently trained classifiers | (e1) | df | cd | pr | model ensemble | (10) | 59.7 | 45.7 |
| | | (e2) | df | cd | pr | weight average | (10) | 57.5 | 44.7 |

descriptions $\mathcal{T}^{\texttt{cd}}$ for $\hat{p}_\theta^{\texttt{t2}}(y|x)$, i.e., $\texttt{t2} = \texttt{cd}$, enhances robust feature preservation and leads to better OOD performance. The other alternative (b2), which employs $\mathcal{T}^{\texttt{cd}}$ for both $\hat{p}_\theta^{\texttt{t1}}(y|x)$ and $\hat{p}_\theta^{\texttt{t2}}(y|x)$, i.e., $\texttt{t1} = \texttt{t2} = \texttt{cd}$, slightly improves (b1). Intriguingly, (b2) is still much worse than (S) despite they both use $\texttt{cd}$ for $\texttt{t2}$.

**(c) Proxy model design:** In our standard setting, $\texttt{type} = \texttt{pr}$. As discussed in Section 4, the proxy term $\hat{p}_{\theta_0}^{\texttt{pr}}(y|x_i)$ is based on the affinity $A_{\theta_0}(x, t_y^{\texttt{cd}}) = \langle f_{\phi_0}(x), g_{\psi_0}(t_y^{\texttt{cd}}) \rangle$ according to the pre-trained CLIP model $\theta_0 = (\phi_0, \psi_0)$ and the set of concept descriptions $\mathcal{T}^{\texttt{cd}}$. In Section 4, we also mentioned the following direct estimation of the oracle model $p_{\text{c}}(y|x)$:

$$\hat{p}_{\theta_0}^{\texttt{cd}}(y|x) = \frac{\exp(A_{\theta_0}(x, t_y^{\texttt{cd}})/\tau)}{\sum_{y' \in \mathcal{Y}} \exp(A_{\theta_0}(x, t_{y'}^{\texttt{cd}})/\tau)}. \tag{15}$$

However, as discussed in Section 4, $\hat{p}_{\theta_0}^{\texttt{cd}}(y|x)$ is susceptible to artifact terms. Consequently, we made a technical adjustment to mitigate the influence of these terms, resulting in the refined $\tilde{p}_{\text{c}}(y|x)$, which is denoted as $\hat{p}_{\theta_0}^{\texttt{pr}}(y|x)$ here. As shown in Table 10, the importance of this adjustment is empirically verified by the much lower performance of (c1) compared to (S).

One can also define $\hat{p}_{\theta_0}^{\texttt{pr-df}}(y|x)$ by replacing $\mathcal{T}^{\texttt{cd}}$ with $\mathcal{T}^{\texttt{df}}$ in the formulation of $\hat{p}_{\theta_0}^{\texttt{pr}}(y|x)$. As shown by the result of (c2), this alternative still underperforms $\hat{p}_{\theta_0}^{\texttt{pr}}(y|x)$ used in the standard setting. This discrepancy can be explained by the fact that the affinities between default text prompts and images are easily affected by changes in the non-core visual features instead of focusing on the core visual features, which has been discussed in Appendix D.

Another simple alternative, denoted by (c3), is to employ the ground-truth one-hot labels as the proxy. Perhaps unsurprisingly, the OOD performance of (c3) is notably inferior to (S) based on the affinities between the images and the concept descriptions.

**(d) Training with either ERM or WRM:** Training with only the first term in (7) results in ERM models (d1) and (d2), whereas training with only the second term leads to a WRM model (d3). Comparing them with DRM models (a1) and (a2), it is clear that models trained to minimize a single risk underperform those trained to minimize both risks, highlighting the importance of dual risk minimization.

**(e) Classifier combination strategy:** Our standard DRM training jointly minimizes the two risks, but one can also train an ERM model $\hat{p}_{\theta_{\text{ERM}}}^{\texttt{df}}(y|x)$ and a WRM model $\hat{p}_{\theta_{\text{WRM}}}^{\texttt{cd}}(y|x)$ separately. These models can be combined for inference using techniques like model ensembling or weight-space averaging. The last two rows of Table 10 show that combining (d1) and (d3) via model ensembling or weight-space averaging generally underperforms joint training (S).

### E.5 Results of applying DRM with FT and LP-FT

Our experiment results reported in Section 5 are based on combining DRM with FLYP. Specifically, we follow FLYP to update both the image encoder and text encoder, and utilize the FLYP loss for the ERM component of (7). The results showed in Table 11 indicate that even without FLYP, where DRM is applied with standard full fine-tuning (Row 2), our method still outperforms FLYP (Row 1), which itself has been shown to surpass standard full fine-tuning in the ERM setting (Goyal et al., 2023). This performance advantage is demonstrated in the results of fine-tuning CLIP ViT-L/14 on iWildCam:

Table 11: Results of applying DRM under different settings.

| Row | FLYP | LP-FT | DRM | ID | OOD |
|-----|------|-------|-----|------|------|
| 1 | Yes | No | No | 56.0 | 41.9 |
| 2 | No | No | Yes | 54.4 | 43.9 |
| 3 | No | Yes | Yes | 56.5 | 46.3 |
| 4 | Yes | No | Yes | 61.8 | 49.2 |

As the table above demonstrates, combining DRM with LP-FT (Row 3) enhances performance over just DRM with standard full fine-tuning (Row 2). Furthermore, integrating DRM with FLYP (Row 4) yields even more significant improvements. Consequently, we adopt the combination of DRM and FLYP as our default setting when fine-tuning CLIP models.

### E.6 Results of applying DRM on ImageNet pre-trained ResNet50

While this work focus on the fine-tuning of zero-shot models that are pre-trained on large-scale image-text pairs, we also explore the possibility of applying DRM on fine-tuning the ImageNet pre-trained CNN models.

When applying DRM to CNN models, we add two randomly initialized classification heads on top of the model. Analogous to the application of DRM on zero-shot models, one classification head is trained using cross-entropy loss with respect to the ground-truth labels, while the other is trained using cross-entropy loss relative to the soft labels generated by the pre-trained zero-shot model. We employ this approach to fine-tune an ImageNet pre-trained ResNet50 model on the iWildCam dataset, utilizing soft labels generated by the CLIP ViT-L/14 model. During inference, the outputs from the two classification heads are combined in a manner similar to that described in (10). The results are presented in Table 12. It is evident from the results that DRM significantly enhances the OOD performance of ResNet50 compared to the ERM.

Table 12: Results of applying DRM on fine-tuning ImageNet pre-trained ResNet50 on iWildCam.

| Method | ID | OOD |
|-----------|------|------|
| ERM+FT | 51.6 | 33.7 |
| ERM+LP-FT | 50.5 | 36.4 |
| DRM+LP-FT | 51.0 | 39.1 |

## F Training details

### F.1 Hyperparameter settings

We primarily adopted the hyperparameter settings from the code released by FLYP (Goyal et al., 2023).

Specifically, for iWildCam, the settings were as follows: `training epochs=20`, `learning rate=1e-5`, `batch-size=256`, and `optimizer=AdamW` with `weight decay=0.2`;

For FMoW, the settings were as follows: `training epochs=20`, `learning rate=1e-5`, `batch-size=256`, and `optimizer=AdamW` with `weight decay=0.2`;

For ImageNet, the settings were as follows: `training epochs=10`, `learning rate=1e-5`, `batch-size=256`, and `optimizer=AdamW` with `weight decay=0.1`.

In all experiments, for the images, we applied the standard CLIP image pre-processing, which included resizing, center cropping, and normalization. For the texts, we applied the standard CLIP text pre-processing, which tokenized the texts into a series of integers, each representing a unique series of characters.

The value of $\lambda$ used in our DRM training was picked from $\{1, 2, 3, 4, 5\}$ based on the performance on the ID validation set. Following Goyal et al. (2023), we also implemented early stopping based on the ID validation performance.

The hyperparameter $\rho$ in WiSE-FT, $\theta_{\text{wise-ft}} = \rho \cdot \theta_{\text{zs}} + (1 - \rho) \cdot \theta_{\text{ft}}$, is chosen from the range 0.1 to 0.9 via ID validation.

## F.2 Machines

All experiments were conducted on a high-performance computing cluster equipped with NVIDIA DGX H800 nodes. Two H800 GPUs with 80 GB VRAM were utilized for all trainings involving CLIP ViT-B/16 and CLIP ViT-L/14, while four H800 GPUs were employed for the training of CLIP ViT-L/14@336.

## F.3 Analysis of computational costs

In our experiments, we implemented the ERM part of DRM with FLYP (Goyal et al., 2023). The additional computational cost of DRM, compared to FLYP, primarily arises from the preparation of soft labels for the targets of the second risk in DRM and the minimization of this second risk. In short, the training and inference cost for DRM increased by about 20% from FLYP. This additional cost is insignificant compared to the attained performance gain. In Table 13, we detail the computational costs and timing for fine-tuning CLIP ViT-L/14 models on ImageNet.

The computation time reported below is based on the setting that training batch size=256 and inference batch size=1024. There are 1,281,167 training images in ImageNet, and thus there are 5005 training batches.

Table 13: Analysis of computational costs for DRM.

| Model | Generating Concept Description by LLM | Soft Label Generation (9) | Training | Inference |
|---|---|---|---|---|
| FLYP | N/A | N/A | In average: **58s/100 batches**, ~48 mins per training epoch | Inference on 100 batches of images takes ~**1.5s**. |
| DRM | We utilized the GPT-4-turbo API to generate concept descriptions for 1,000 ImageNet classes, inputting 10 classes at a time to ensure quality. The generation cost is **under 10 US Dollar** (the price of the API is 10 US Dollar/1 million prompt tokens). We are unaware of the computational cost as the model details of GPT-4 are unknown. | The primary computational cost arises from using pre-trained CLIP models to generate image and text embeddings from 1,281,167 training images and 1,000 concept descriptions. Soft labels are created using inner products between these embeddings, with some technical adjustments. The entire process takes **less than 3 minutes**. | In average: **71s/100 batches**, ~58 mins per training epoch | Inference on 100 batches of images takes ~**1.8s**. |

## G  Limitations

Our research utilized GPT-4 to generate concept descriptions for the core visual features of various classes across different domains. It is important to note that the scope of GPT-4's knowledge in certain domains might be limited, and as a result, the model may not always generate useful concept descriptions. For instance, we found that GPT-4 generated inaccurate concept descriptions in medical imaging fields like ocular disease and breast histology.

Additionally, due to the vast number of concept descriptions generated, we have not been able to verify the accuracy of each generated concept description. To enhance the quality of these descriptions, potential improvements could involve engaging domain experts to review and correct errors, or generating descriptions manually. Another approach could be to gather visual prototypes and use advanced multimodal LLMs such as GPT-4V (Achiam et al., 2023), LLaVA (Liu et al., 2023), or MiniGPT-4 (Zhu et al., 2024), which might yield more precise descriptions of the core visual features.

A further limitation concerns the CLIP models used in our experiments. These models may not perform optimally across all domains, particularly in less common areas, where they may lack requisite knowledge in both images and text. The effectiveness of our DRM method is therefore contingent upon the breadth and depth of the pre-training data of CLIP models. Unfortunately, the specifics of the CLIP pre-training dataset have not been disclosed by OpenAI, adding an element of uncertainty to the performance of our method in niche domains.

