# OpenReview forum: "Dual Risk Minimization: Towards Next-Level Robustness in Fine-tuning Zero-Shot Models"
_NeurIPS.cc/2024/Conference — NeurIPS 2024 poster_

### Official Review · Reviewer_gi7f · 2024-07-05

**Soundness:** 2
**Presentation:** 2
**Contribution:** 2
**Rating:** 6
**Confidence:** 3

**Summary:**

To address the robustness of foundational models under distribution shift conditions, this paper proposes a dual risk minimization approach. Specifically, the authors combine Empirical Risk Minimization (ERM) and Worst-Case Risk Minimization (WCRM) to optimize the model fine-tuning process. To achieve accurate distribution estimation, the authors use class descriptions generated by GPT-4 as prompts inputted into the model and apply max-min normalization to the classification probabilities to address certain items that should be zero. Experiments demonstrate that the dual risk minimization method significantly enhances robustness in zero-shot fine-tuning.

**Strengths:**

1. The authors propose dual risk minimization through rigorous mathematical proofs.
2. Experiments show that dual risk minimization can effectively enhance the robustness of the model.

**Weaknesses:**

1. To accurately estimate $p_c(y|c) $, the authors use prompts generated by GPT-4 to initialize the classifier parameters. Nevertheless, this classifier may still not be the optimal choice.
2. Given that better prompts have been used to initialize the classifier cd, why not directly fine-tune cd instead of using cd to assist in fine-tuning df? Moreover, why is the performance of cd inferior to df when comparing row 2 and row 3 in Table 3?
3. In Table 7, if the models trained with df and cd (row d1 and d2) are directly used for inference with (13), how will the performance be?

**Questions:**

See Weakness.

**Limitations:**

Yes

---

> ### Author Rebuttal · Authors · 2024-08-07
>
> We appreciate the reviewer's time and valuable insights. It is encouraging that the reviewer found our DRM approach is supported by "rigorous mathematical proofs" and it “effectively enhances the robustness of the model”. We appreciate that the reviewer has thoroughly engaged with our work and acknowledged its merits.
>
> Regarding the concerns of the reviewer, we provide detailed clarifications below.
>
> ---
>
> ### 1. Accuracy of the estimate of $p_c(y|x)$
>
> There is certainly still a gap between our estimate and the real $p_c(y|x)$. However, we would like to highlight that the estimate is already substantially better than one-hot labels, as demonstrated by the comparison between (S) and (c3) in Table 7: OOD performance of 49.2 vs. 45.1. Moreover, our approach requires minimal human input, making it highly scalable. In principle, with more human input, we should be able to further improve the estimate, and this would be an interesting future direction.
>
> ---
>
> ### 2. “... why not directly fine-tune cd instead of using cd to assist in fine-tuning df?”
>
> Please see **B. Why not directly fine-tune a concept-description classifier?** in the global response. There, we show experiment results and discuss why solely fine-tuning the concept-description classifier is not better than our DRM fine-tuning.
>
> ---
>
> ### 3. “... why is the performance of cd inferior to df when comparing row 2 and row 3 in Table 3?”
>
> Please note that only the ID performance of cd is inferior to df. This is expected since cd is not (while df is) trained with ERM which aims to maximize ID performance.
>
> ---
>
> ### 4. “In Table 7, if the models trained with df and cd (rows d1 and d2) are directly used for inference with (13), how will the performance be?”
>
> This is an interesting question. Upon further experiments, we find that: 1. Using a mixture model to make predictions for (d1), which is trained with default prompts (df), results in slightly improved OOD performance, though it still significantly lags behind that of the full DRM approach; 2. Conversely, using a mixture model for predictions in (d2), which is trained with concept descriptions (cd), actually leads to a decrease in OOD performance.
>
> |Model|t1|t2|Inference|ID|OOD|
> |--|--|--|--|--|--|
> |DRM|df|cd|mixture|61.8|49.2|
> |(d1)|df|/|df|56.0|41.9|
> ||||cd|10.1|8.8|
> ||||mixture|56.4|42.4|
> |(d2)|cd|/|cd|56.9|43.4|
> ||||df|12.3|9.4|
> ||||mixture|56.3|42.7|
>
> Recall that, (d1) refers to the ERM-trained model that uses a concentration of text embeddings from default prompts as the classification head, while (d2) represents the ERM-trained model utilizing a concentration of text embeddings from concept descriptions as the classification head. For both (d1) and (d2), the mixture hyperparameter $\beta$ is determined based on achieving the best ID validation performance.

---

> > ### Comment · Reviewer_gi7f · 2024-08-07
> >
> > The author have addressed my concern, and I will change my score from 5 to 6.

---

> > > ### Author Response · Authors · 2024-08-09
> > >
> > > Thank you for taking the time to read our response and increasing your score! We are glad to hear that the response addressed your concerns.

---

### Official Review · Reviewer_zSx8 · 2024-07-14

**Soundness:** 4
**Presentation:** 3
**Contribution:** 4
**Rating:** 5
**Confidence:** 4

**Summary:**

This paper proposes a method for robust fine-tuning by combining empirical risk minimization with worst-case risk minimization to better preserve core features. The approach uses descriptions of core features obtained from large language models (LLMs) like GPT-4 and employs these descriptions to estimate worst-case risk. This method aims to enhance robustness by minimizing empirical risk on the training set of downstream tasks while improving the understanding of core features. As a result, the proposed method demonstrates significant performance improvements over existing methods on datasets such as ImageNet, iWildCam, and FMoW.

**Strengths:**

- The paper introduces a novel approach by combining empirical risk minimization with worst-case risk minimization, particularly when no other domain is provided. The use of LLMs to obtain core-feature descriptions is innovative and allows for practical implementation without human annotations. Additionally, integrating this method with techniques like weight ensemble can lead to synergistic performance improvements.
- The method is well-supported by theoretical foundations and empirical results. The significant performance improvements on challenging datasets like iWildCam and FMoW highlight the robustness and effectiveness of the approach.
- The paper clearly explains the motivation behind combining empirical and worst-case risk minimization, and the experimental results support the claimed performance improvements. Table 3 effectively highlights the necessity of redefining the new proxy oracle due to artifact terms.

**Weaknesses:**

- The main weakness of the proposed method lies in the sensitivity to the hyperparameter $\lambda$ . As shown in Table 6, slight deviations from the optimal $\lambda$  value can lead to significant performance drops. Finding the optimal $\lambda$  for different datasets could be impractical, and this issue needs to be addressed by reporting ablation studies on additional datasets beyond iWildCam.
- The results in Table 6 show no trade-off between in-distribution (ID) and out-of-distribution (OOD) performance, which is counterintuitive. The authors should provide an explanation for this observation.
- While the overall quality of writing is high, certain sections and figures need improvement. For example:
	- Line 159 and Eq. 3 need a clearer explanation of the DRM formulation relaxation process.
	- Figure 1 should explicitly state the meaning of the first and second bars.
	- Figure 2 could be misleading as it shows the original image having higher affinity with the class concept description than the background-free image without showing affinities with other images.
- The method relies heavily on the quality of the class concept descriptions generated by GPT-4. An analysis and ablation study on the quality of these descriptions are necessary. Additionally, the impact of using different LLMs should be explored.

**Questions:**

- The proposed method has only been tested with full fine-tuning. I think this method also can be combined with methods like LP-FT or FLYP. I am curious about the results when they are combined with the proposed method.
- Why is there no observed trade-off between ID and OOD performance in Table 6? An explanation for this phenomenon is needed.
- How robust is the method to the quality of class concept descriptions? What happens when a different LLM is used?

**Limitations:**

The authors have addressed some limitations of their work but have not sufficiently discussed the critical issue of hyperparameter sensitivity and its impact on performance. There are no identified negative societal impacts.

---

> ### Author Rebuttal · Authors · 2024-08-07
>
> We appreciate the reviewer's time and valuable insights. It is encouraging that the reviewer found our DRM approach "novel", the use of LLMs to obtain core-feature descriptions is "innovative", and the method is "well-supported by theoretical foundations and empirical results". We appreciate that the reviewer has thoroughly engaged with our work and acknowledged its merits.
>
> Regarding the concerns of the reviewer, we provide detailed clarifications below.
>
> ---
>
> ### 1. Impact of hyperparameter $\lambda$ in DRM
>
> $\lambda$ is a balancing hyperparameter for the two risks in DRM.  While it may seem that DRM is sensitive to $\lambda$, we would like to draw your attention to the fact that the range of $\lambda$ with decent performance is still wide. As shown in Table 6 (which we copy below), for $\lambda$ between 2 and 5, DRM maintains high-level OOD performance (above 48.1, compared to FLYP’s 42.1) on iWildCam. In other words, the performance drop (from 49.2 to 48.1) is relatively small compared to the gain.
>
> **Copy of Table 6**
>
> |$\lambda$|ID|OOD|
> |--|--|--|
> |0 (FLYP)|56.0|41.9|
> |0.1|56.4|42.6|
> |0.5|57.2|43.9|
> |1|59.1|47.3|
> |2|60.0|48.1|
> |3|61.8|49.2|
> |4|60.9|48.6|
> |5|60.1|48.5|
> |10|55.4|47.7|
> |50|52.5|46.6|
>
> Following your great suggestion, we have conducted the same experiments on ImageNet with CLIP ViT-B/16 to further study the impact of $\lambda$:
>
> |$\lambda$|ID|OOD|
> |--|--|--|
> |0 (FLYP)|82.6|60.2|
> |1|81.5|62.5|
> |2|81.8|63.1|
> |3|82.0|63.2|
> |4|81.9|63.4|
> |5|81.7|63.3|
> |6|81.8|63.2|
> |20|81.5|62.3|
> |50|81.2|61.9|
>
> Similar to the results on iWildCam, DRM achieves great performance across a wide range of $\lambda$ between 2 and 6. All in all, these results suggest that DRM is fairly insensitive to the choice of $\lambda$ as it is easy to find appropriate values of $\lambda$ for DRM to significantly outperform the state of the art.
>
> ---
>
> ### 2. “The results in Table 6 show no trade-off between ID and OOD performance, which is counterintuitive ...”
>
> Indeed, this was also counterintuitive to us at first, but then a closer inspection revealed a rather intuitive explanation. The ID performance is determined by two factors: (i) how well the model fits the training data, and (ii) how severe the overfitting (if any) is. The results in Table 6 show no trade-off because for relatively small $\lambda$, (i) the model can still fit the training data reasonably well, and (ii) the WRM objective can in fact reduce overfitting, as shown by the training/validation performance of DRM on iWildCam below:
>
> |$\lambda$|Training Acc|ID Val Acc|
> |--|--|--|
> |0|92.37|79.44|
> |1|88.29|81.64|
> |3|87.41|82.43|
>
> This explains why ID performance improves as $\lambda$ increases (for $\lambda \leq 3$), and consequently why no trade-off is observed in Table 6.
>
> ---
>
> ### 3. Writing
>
> Thank you for your great suggestions.
>
> - We will make the explanation of the DRM formulation relaxation process clearer by showing the key intermediate steps from Eq. (3) to line 159.
> - We will specify the meaning of the bars in Figure 1: the 1st bar – the predicted probability of skis being present in the image; and the 2nd bar - no skis being present in the image.
> - To avoid potential misunderstanding, we will add the affinities w.r.t. other images for Figure 2. We provide several such examples in Figure 2 of the attached PDF in the global response.
>
> ---
>
> ### 4. Quality of concept descriptions and its impact on the results
>
> Please see **A. Quality and Robustness of LLM-Generated Concept Descriptions** in the global response. There, we have included the results of a quantitative study on the reliability of concept descriptions, an analysis of the stochastic variability in LLM generations, and an investigation into the robustness of DRM with respect to concept descriptions generated by various LLMs.
>
> In summary, the results demonstrate that GPT-4's concept descriptions effectively capture the core features, with the resulting affinities proving stable and unaffected by randomness in the LLM generation process, and DRM is robust to the concept descriptions generated by different LLMs.
>
> ---
>
> ### 5. Combining DRM with LP-FT or FLYP
>
> Yes, our DRM can indeed be integrated with other methods such as LP-FT and FLYP. In fact, we have combined DRM with FLYP in our experiments, as detailed in lines 276-278 of our paper. Specifically, we utilized FLYP for the ERM component of Eq. (12). It is crucial to highlight that even without FLYP, where DRM is applied with standard full fine-tuning (row 2), our method still outperforms FLYP (row 1), which itself has been shown to surpass standard full fine-tuning in the ERM setting [1]. This performance advantage is demonstrated in the results of fine-tuning CLIP ViT-L/14 on iWildCam:
>
> |Row|FLYP|LP-FT|DRM|ID|OOD|
> |--|--|--|--|--|--|
> |1|Yes|No|No|56.0|41.9|
> |2|No|No|Yes|54.4|43.9|
> |3|No|Yes|Yes|56.5|46.3|
> |4|Yes|No|Yes|61.8|49.2|
>
> As the table above demonstrates, combining DRM with LP-FT (row 3) enhances performance over just DRM with standard full fine-tuning (row 2). Furthermore, integrating DRM with FLYP (row 4), as we have explored in our paper, yields even more significant improvements.
>
> In the next version of our paper, we will further clarify that our primary experimental configurations involve DRM combined with FLYP, and will include the above discussions you kindly suggested.
>
> ---
>
> **References:**
>
> [1] Finetune like you pretrain: Improved finetuning of zero-shot vision models. CVPR, 2023.

---

> > ### Comment · Reviewer_zSx8 · 2024-08-14
> >
> > Thank you for your thorough response. I have carefully reviewed all the rebuttals and your clarifications, which have addressed many of my concerns. However, some issues still remain.
> >
> > First, the observation that increasing $\lambda$ leads to improvements in both ID and OOD performance appears to be specific to the iWildCam dataset. The results on ImageNet show a different trend. Although it is true that $\lambda$ values between 2 and 5 generally result in good ID and OOD performance, the consistent improvement in OOD performance as ID performance increases does not seem to hold universally.
> >
> > Moreover, all ablation studies have been conducted solely on iWildCam, whereas the qualitative examples are shown on different datasets. This narrow focus on a single dataset for ablation and analysis makes it difficult to fully understand and validate the proposed method. I believe it is necessary to provide experimental results and analyses across a wider variety of datasets to ensure a comprehensive understanding of the proposed approach.
> >
> > The concept of combining ERM and WRM is intriguing, and the approach is interesting. However, the need to generate concept descriptions for all classes, along with the fact that the effectiveness of this approach varies depending on the quality of the generated descriptions, the $\lambda$ hyperparameter reduces the practical utility of this method. For these reasons, I intend to maintain my current score.

---

### Official Review · Reviewer_eddC · 2024-07-22

**Soundness:** 3
**Presentation:** 3
**Contribution:** 3
**Rating:** 6
**Confidence:** 2

**Summary:**

This paper introduces dual risk minimization (DRM), a novel approach that combines empirical risk minimization (ERM) with worst-case risk minimization (WRM) to enhance the robustness of fine-tuning zero-shot foundation models. The authors address the limitations of existing methods that fail to effectively preserve robust features during fine-tuning by utilizing concept descriptions from LLMs to create soft labels for estimating worst-case risk, focusing on core features that define target classes. Empirical results demonstrate that DRM significantly improves out-of-distribution performance on benchmarks such as ImageNet and WILDS, establishing new state-of-the-art results.

**Strengths:**

- The main idea of dual risk minimization (DRM), which combines ERM and WRM, is novel and interesting. Also, DRM effectively balances expected and worst-case performance.
- Experimental results show that DRM achieves state-of-the-art results on various benchmarks.

**Weaknesses:**

- The dependence on the lambda seems to be quite significant, and there is no ablation study addressing this in the main paper. It would be good to see results regarding the impact of lambda.
- It would be helpful if the paper included details on the computational time and cost for each experiment to understand the efficiency and scalability of the proposed methods.

**Questions:**

Please refer to the weaknesses.

**Limitations:**

Yes. They address the limitations of their work.

---

> ### Author Rebuttal · Authors · 2024-08-07
>
> We are grateful for the reviewer's time and insightful feedback. It is encouraging to know that the reviewer found our DRM “novel and interesting” and “achieves state-of-the-art results on various benchmarks”. We appreciate that the reviewer has thoroughly engaged with our work and acknowledged its merits.
>
> Regarding the concerns of the reviewer, we provide detailed clarifications below.
>
> ---
>
> ### 1. Impact of hyperparameter $\lambda$ in DRM
>
> $\lambda$ is a balancing hyperparameter for the two risks in DRM. We have discussed its impact on the performance of DRM in Appendix D.3 of our paper. Here, we copy Table 6 from Appendix D.3 below. Notably, for $\lambda$ within the wide range from 2 to 5, DRM maintains high-level OOD performance (above 48.1, compared to FLYP’s 42.1) on iWildCam.
>
> **Copy of Table 6**
> |$\lambda$|ID|OOD|
> |--|--|--|
> |0 (FLYP)|56.0|41.9|
> |0.1|56.4|42.6|
> |0.5|57.2|43.9|
> |1|59.1|47.3|
> |2|60.0|48.1|
> |3|61.8|49.2|
> |4|60.9|48.6|
> |5|60.1|48.5|
> |10|55.4|47.7|
> |50|52.5|46.6|
>
> To further study the impact of $\lambda$, we have also conducted the same experiments on ImageNet with CLIP ViT-B/16:
>
> |$\lambda$|ID|OOD|
> |--|--|--|
> |0 (FLYP)|82.6|60.2|
> |1|81.5|62.5|
> |2|81.8|63.1|
> |3|82.0|63.2|
> |4|81.9|63.4|
> |5|81.7|63.3|
> |6|81.8|63.2|
> |20|81.5|62.3|
> |50|81.2|61.9|
>
> Similar to the results on iWildCam, DRM achieves great performance across a wide range of $\lambda$ between 2 and 6. All in all, these results suggest that DRM is fairly insensitive to the choice of $\lambda$ as it is easy to find appropriate values of $\lambda$ for DRM to significantly outperform the state of the art.
>
> ---
>
> ### 2. Computation time and cost
>
> In our experiments, we implemented the ERM part of DRM with FLYP [1]. The additional computational cost of DRM, compared to FLYP, primarily arises from the preparation of soft labels for the targets of the second risk in DRM and the minimization of this second risk. **In short, the training and inference cost for DRM increased by about 20% from FLYP. This additional cost is insignificant compared to the attained performance gain.** Below, we detail the computational costs and timing for fine-tuning CLIP ViT-L/14 models on ImageNet.
>
> We use two Nvidia H800 GPUs with 80GB VRAM (a modified version of the Nvidia H100 with a reduced chip-to-chip data transfer rate) and 24 CPU cores from Intel Xeon Scalable processors. The computation time reported below is based on the setting that training batch size=256 and inference batch size=1024. There are 1,281,167 training images in ImageNet, and thus there are 5005 training batches.
>
> | Model | Generating Concept Description by LLM | Soft Label Generation - Eq. (11) | Training | Inference |
> |---|-------|----------------------------------|----------|-----------|
> | FLYP  | N/A                                   | N/A                              | In average: **58s/100 batches**, ~48 mins per training epoch | Inference on 100 batches of images takes **~1.5s**. |
> | DRM   | We utilized the GPT-4-turbo API to generate concept descriptions for 1,000 ImageNet classes, inputting 10 classes at a time to ensure quality. The generation cost is **under 10 US Dollar** (the price of the API is 10 US Dollar/1 million prompt tokens). We are unaware of the computation cost as the model details of GPT-4 are unknown. | The primary computational cost arises from using pre-trained CLIP models to generate image and text embeddings from 1,281,167 training images and 1,000 concept descriptions. Soft labels are created using inner products between these embeddings, with some technical adjustments. The entire process takes **less than 3 minutes**. | In average: **71s/100 batches**, ~58 mins per training epoch | Inference on 100 batches of images takes **~1.8s**. |
>
> This detailed analysis of the computation cost and time will be included in the next version of this paper.
>
> ---
>
> **References:**
>
> [1] Finetune like you pretrain: Improved finetuning of zero-shot vision models. CVPR, 2023.

---

### Official Review · Reviewer_D5j1 · 2024-07-25

**Soundness:** 2
**Presentation:** 3
**Contribution:** 2
**Rating:** 5
**Confidence:** 3

**Summary:**

This paper presents Dual Risk Minimization (DRM), a novel approach that combines empirical risk minimization (ERM) and worst-case risk minimization (WRM) for fine-tuning the CLIP model while maintaining its out-of-distribution robustness. The idea is to create a classifier that utilizes concept descriptions of each class generated by large language models (LLMs). This classifier helps “pull out” core features from the image embeddings, serving as a regularizer to enhance the robustness of CLIP fine-tuning. Moreover, the paper introduces a min-max normalization technique to address a caveat in the regularization. Experimental results demonstrate the effectiveness of the proposed DRM, yielding promising in-distribution and out-of-distribution accuracy.

**Strengths:**

- The paper is well-written and generally easy to follow.
- The paper brings an interesting viewpoint for CLIP fine-tuning that separates the image features into core (content) and non-core (style) features.
- The experimental results look promising, especially on the two specialized datasets, iWILDCam and FMoW.

**Weaknesses:**

- At Line 199, the paper claims that the LLM-generated concept descriptions can “pull out” the core features from the image embeddings. However, this claim is only demonstrated by the few examples shown in Figure 2, which is insufficient. Moreover, the stochastic nature of LLM generation can further complicate the matter. The paper lacks a thorough analysis of the reliability and robustness of the core features obtained through this proposed method.

- At Line 217, the paper mentions that the affinity between $x$ and any other class $y'$ should ideally be zero, thereby motivating the modification of the second term in Eq. 10. However, this might ignore the fact that the ground-truth class $y$ exhibits varying degrees of similarity with different $y’$. It remains unclear how the proposed method can achieve better performance, as shown in Table 3, while disregarding the learning of such inter-class similarities.

- The second term in Eq. 10. is similar to knowledge distillation, as acknowledged by the paper at Line 207. However, the paper neglects to discuss or compare this term with other robust fine-tuning methods, such as [1], which also incorporate knowledge distillation.

[1] DELTA: Deep Learning Transfer using Feature Map with Attention for Convolutional Networks, ICLR 2019.

**Questions:**

Besides the weakness shown in the above section, please also see the following questions:

- Given that the paper suggests that the classifiers built with LLM-generated concept descriptions have the ability to extract core features, could we simply fine-tune the CLIP model utilizing these concept descriptions instead of using the one with default descriptions? If we do so, could standard fine-tuning with ERM already maintain its robustness?

- What would be the in-distribution and out-of-distribution accuracy for the concept description classifiers? Given that these classifiers aim to extract more core features, would they be more robust to distribution changes, even without fine-tuning?

- How does the min-max normalization affect the training on long-tailed datasets, like iWILDCam? Specifically, since the min-max normalization computes $x$ with all images in a class $y$, could tailed classes result in less stable normalization because they have much fewer samples?

**Limitations:**

The paper acknowledges that a potential limitation of the proposed method lies in the potential limited domain knowledge of LLMs in specific domains. To better illustrate this limitation, it would be beneficial for the paper to provide concrete examples, such as failure cases, that show how the method behaves when it reaches its limits.

---

> ### Author Rebuttal · Authors · 2024-08-07
>
> We are grateful for the reviewer's time and insightful feedback. Regarding the concerns and questions the reviewer raised, we provide detailed clarifications below. If not specified otherwise, all experiments below are conducted with CLIP ViT-B/16.
>
> ---
> ### 1. Reliability of the core features obtained with concept descriptions
>
> Please see **A. Quality and Robustness of LLM-Generated Concept Descriptions** in the global response.
>
> ---
>
> ### 2. “At Line 217, the paper mentions that the affinity between $x$ and any other class $y'$ should ideally be zero ... this might ignore the fact that the ground-truth class $y$ exhibits varying degrees of similarity with different $y'$ ...”
>
> This is not what we stated in the paper. At lines 217-218, please note that “$x$ is *completely void* of core visual features for $y'$” and only in such case the affinities should ideally be 0.
>
> It is generally true that, as you mentioned, “the ground-truth class $y$ exhibits varying degrees of similarity with different $y’$”. Our method does *not* disregard the learning of such inter-class similarities; on the contrary, our method specifically captures the similarities with Eq. (11) where the probabilities of non-ground-truth classes are weighted by their affinities w.r.t. the image.
>
> To illustrate this point, for each class in ImageNet, we identify the classes with the highest average probabilities in the soft labels generated by Eq. (11). Below are some examples:
>
> - horizontal bar: horizontal bar, balance beam, parallel bars, pole, swing;
> - whiskey jug: whiskey jug, water jug, pitcher, beer bottle, beaker;
> - goldfish: goldfish, rock beauty, tench, barracouta, anemone fish.
>
> We can see that the top-1 class is consistently the class itself as intended (in fact, this holds for all 1k ImageNet classes), and the other top classes are indeed visually related to the top-1 class.
>
> ---
>
> ### 3. Comparison with knowledge distillation methods
>
> Yes, the term is related to knowledge distillation, or more precisely, self-distillation. There is a self-distillation method, L2-SP, in Table 1 of our paper. Here, we compare our method with two more state-of-the-art self-distillation methods, CaRot [1] and MIRO [2]. Their ID/OOD performances are shown in the table below.
>
> ||ImageNet |iWildCam |FMoW |
> |--|--|--|--|
> |FLYP|82.6/60.2|52.2/35.6|68.6/41.3|
> |CaRot|83.1/62.5|49.8/34.3|68.8/39.8|
> |MIRO|----/----|51.6/37.2|66.1/42.2|
> |DRM|82.0/63.2|54.1/40.0|68.7/45.9|
>
> We will further discuss related methods in the next version of the paper, including CaRot, MIRO, and DELTA [3] which you kindly suggested. Please note that DELTA has been surpassed by MIRO [2] on various benchmarks so we only included MIRO here.
>
> ---
>
> ### 4. “... could we simply fine-tune the CLIP model utilizing these concept descriptions? ...”
>
> Please see **B. Why not directly fine-tune a concept-description classifier?** in the global response.
>
> ---
>
> ### 5. “What would be the ID and OOD accuracy for the concept description classifiers? ... would they be more robust to distribution changes, even without fine-tuning?”
>
> As shown below, they are slightly better than the default-prompt classifiers, but much worse than the fine-tuned DRM classifiers. In short, concept descriptions are helpful, but fine-tuning is still important, especially when the domain gap is large, e.g., in the cases of iWildCam and FMoW.
>
> |Model|ImageNet |iWildCam |FMoW |
> |--|--|--|--|
> |Zero-shot w/ default prompts|68.3/58.7|8.7/11.0|20.4/18.7|
> |Zero-shot w/ concept descriptions|68.6/59.0|11.49/12.84|20.6/19.8|
> |DRM|82.0/63.2|54.1/40.0|68.7/45.8|
>
> ---
>
> ### 6. “How does the min-max normalization affect the training on long-tailed datasets ...?”
>
> This is a great question. We concur that using min-max normalization on long-tailed classes can lead to less stable normalization. That said, its impact on overall performance should be small.
>
> The reason is two-fold. First, even if there are only a few samples for a class, most samples of the class would still have higher affinities with the class compared to samples from other classes. This means that $\gamma(x, y)$ would be small for $y \neq y_x$, and thus any impact on the estimation of $p_c(y|x)$ (via Eq. (11)) would likely also be small. Second, for the estimation of $p_c(y_x|x)$, although with fewer samples the difference in $p_c(y_x|x)$ between $x$ of the same class $y_x$ would likely be magnified, their relative ordering would still be intact, and thus the classifier would still learn to preserve the core features of the class.
>
> Surprisingly, we find that DRM can even enhance the learning of long-tailed classes on iWildCam. This is reflected by the table below where we can see the training F1 score increases while the training accuracy decreases as $\lambda$ increases.
>
> | $\lambda$ | training acc | training F1 score | ID val acc | ID val F1 score |
> |--|--|--|--|--|
> | 0 | 92.37 | 67.72 | 79.44 | 46.64 |
> | 1 | 88.29 | 78.96 | 81.64 | 51.36 |
> | 3 | 87.41 | 79.44 | 82.43 | 52.68 |
>
> In particular, for classes with fewer than 50 training examples, we find that FLYP achieves an accuracy of 59.61% on the ID validation set, whereas DRM achieves a significantly higher accuracy of 69.38%.
>
> ---
>
> ### 7. About limitations
>
> Thank you for suggesting the use of specific examples to illustrate DRM limitations due to LLMs' restricted domain knowledge. In our next version, we will showcase GPT-4's inaccuracies in medical imaging fields like ocular disease and breast histology.
>
> ---
>
> **References:**
>
> [1] Towards calibrated robust fine-tuning of vision-language models. arXiv, 2023.
> [2] Domain generalization by mutual-information regularization with pre-trained models. ECCV, 2022.
> [3] DELTA: Deep Learning Transfer using Feature Map with Attention for Convolutional Networks, ICLR 2019.

---

### Author Rebuttal · Authors · 2024-08-07

We thank all reviewers for their meticulous reviews of our work. In this global response, we address two main concerns shared by some reviewers.

## A. Quality and Robustness of LLM-Generated Concept Descriptions

Reviewers D5j1 and zSx8 stressed the importance of thoroughly analyzing concept descriptions generated by LLMs. D5j1 focused on the reliability and robustness of these descriptions, while zSx8 recommended an ablation study to assess their quality and the effects of using various LLMs. Following their kind suggestions, we conducted further empirical study.

### A.1. Quantitative study on the reliability of concept descriptions

In Figure 2 of our paper, we have showcased two examples that “LLM-generated concept descriptions have the ability to extract core features”. To further support this claim, we present a full quantitative study. The study is conducted with Hard ImageNet [1] and consists of two parts. First, we follow the setting described in Appendix C, i.e., removing image background (BG), and observing how the image-text affinities change for default prompts (df) and concept descriptions (cd) respectively. In the second part, we do the same but with foreground (FG) removed (see Figure 1 of the attached PDF for examples).

The following table shows the average affinities over all 19,097 images across all 15 classes of Hard ImageNet. The percentages in the table indicate the relative changes w.r.t. the affinities of the original images (FG & BG).

||FG & BG|w/o BG|w/o FG|
|-|-|-|-|
|df|0.3473|0.2393 (-31.1%)|0.3407 (-1.9%)|
|cd|0.2660|0.2387 (-10.3%)|0.1180 (-55.6%)|

The result shows that **the affinities of concept descriptions are much more invariant to changes in non-core features than default prompts (-10.3% vs. -31.1%)**. This is consistent with Figure 2 and other examples in Appendix C. Moreover, **the affinities are quite responsive (-55.6%) to changes in core features**. In contrast, the affinities of default prompts barely change (-1.9%) in response to the absence of core features. **These results suggest that concept descriptions are indeed pulling out the core features.** For detailed results of each class, please see Table 1 & 2 in the PDF.

### A.2. On the stochasticity of LLM generations

To evaluate the impact of the stochasticity of LLM generations, we repeatedly ask GPT-4 to generate concept descriptions for each class three times and compute the standard deviation of the resulting image-text affinities. As examples, for class “white-lipped peccary”, the generated descriptions are as follows:
   - Compact, dark grey body with distinctive white markings around the mouth.
   - Compact, dark gray body with distinctive white markings around the lips.
   - A stocky body with coarse, dark hair and distinct white markings around the mouth.

The average standard deviation of the corresponding affinities over 20k randomly sampled images of iWildCam is 0.0061, which is surprisingly small compared to the mean, 0.2659. This shows that **the affinities are quite stable and insensitive to randomness in the generation process**.

### A.3. On the robustness of DRM w.r.t. concept descriptions by different LLMs

We have also experimented with different LLMs of various sizes (from 8B to over 405B parameters) to generate concept descriptions.

|Method|LLM (#params)|ID|OOD|
|-|-|-|-|
|FLYP|-|52.2|35.6|
|DRM|GPT-3.5 (~20B?)|53.4|38.7|
|DRM|GPT-4 (>1T?)|54.1|40.0|
|DRM|Llama-3 (8B)|53.8|39.2|
|DRM|Llama-3 (70B)|54.0|39.9|
|DRM|Llama-3 (405B)|53.9|40.5|

All the LLMs greatly improve baseline performance on iWildCam. **This shows our method is not sensitive to the quality of the concept descriptions either.**

## B. Why not directly fine-tune a concept-description classifier?

Reviewer D5j1 raised a question about the feasibility of directly fine-tuning the CLIP model using concept descriptions (cd). Meanwhile, reviewer gi7f questioned why not directly fine-tune the cd classifier but using it to assist in the default-prompt (df) classifier. These are both great questions.

In fact, we have tried to directly fine-tune the cd classifier. The results on iWildCam have been reported in (b2), (d2) and (d3) of Table 7 in the paper. In (b2), the cd classifier is fine-tuned with both ERM and WRM. In (d2), it is fine-tuned with only ERM; whereas in (d3), it is fine-tuned with only WRM. We copy the results here and provide more explanations.

|||ID|OOD|
|-|-|-|-|
|(b2)|cd classifier (ERM+WRM)|54.0|46.1|
|(d2)|cd classifier (ERM)|56.9|43.4|
|(d3)|cd classifier (WRM)|51.7|46.3|
|DRM|df classifier (ERM) + cd classifier (WRM)|61.8|49.2|

**Both (b2) and (d2) involve ERM in fine-tuning the cd classifier. Notably, they both underperform DRM.** This is because the good performance of DRM does not only come from the cd classifier, but also the soft labels constructed from the concept descriptions (via Eq.(11)). These soft labels are the targets for WRM, a crucial component of DRM besides ERM.

**The problem with ERM in (b2) and (d2) is that it uses one-hot labels which do not capture subtle differences of core visual features among images.** While pre-trained cd classifiers do have the ability to extract core features, fine-tuning them with ERM would certainly strip some of this ability away. In comparison, the df classifier is more aligned with the ERM objective. **In DRM, we therefore separate ERM and WRM for the two classifiers, reducing the interference of ERM in fine-tuning the cd classifier which aims to extract the core features and achieve WRM.**

Finally, comparing (d2) and (d3), we observe that the (d3) cd classifier, fine-tuned with our proposed core-feature-aware soft labels, shows much better OOD performance than the (d2) classifier, which is fine-tuned with one-hot labels. This result clearly demonstrates the advantage of using the soft labels.

**Reference:**

[1] Hard imagenet: Segmentations for objects with strong spurious cues. NeurIPS, 2022.

---

### Author Response · Authors · 2024-08-13
**Message to Reviewers: We look forward to hearing from you**

Dear Reviewers,

Thank you again for reviewing our paper. We are more than happy to address any further questions or concerns you may have before the conclusion of the author-reviewer discussion period (in 30 hours). Please feel free to reach out with any further feedback. We look forward to hearing from you.

Best regards,

Authors

---

### Decision · Program_Chairs · 2024-09-25

**Decision:**

Accept (poster)

**Comment:**

This paper presents a new technique based on a combination of empirical risk minimization and worst-case risk minimization for robust finetuning of zero-shot models. The reviewers appreciated the novel and interesting motivation, theoretical foundation, and strong results of the proposed method as well as the clarity of the manuscript. They also raised concerns with lack of in-depth analysis (e.g., reliability and robustness of the core features obtained by the method, complexity analysis, impact of the quality of generated descriptions and that of the LLM used) (D5j1, eddC, zSx8), sensitivity to the hyperparameter $\lambda$ (zSx8), potential risk of using imperfect estimate of the oracle model (gi7f) and that of ignoring inter-class similarities (D5j1), lack of discussion on counterintuitive results (zSx8), and missing comparisons with closely related work (D5j1). Most of these concerns were successfully addressed by the rebuttal and subsequent responses in the discussion. Consequently, the reviewers unanimously championed the paper after the discussion period. The AC agrees with the reviewers and recommends acceptance. The authors are strongly encouraged to carefully revise the paper to reflect the valuable comments by the reviewers (in particular those on the sensitivity to $\lambda$), to provide a tighter connection between the theorem and the actual DRM objective and its implementation, and to add new results and discussions brought up in the rebuttal and discussion.